# Analyzing Fine-Grained Alignment and Enhancing Vision Understanding in Multimodal Language Models

**Jiachen Jiang**[†], **Jinxin Zhou**[†], **Bo Peng**[†], **Xia Ning**[†,◇,♡], **Zhihui Zhu**[† *]

[†]Department of Computer Science and Engineering, The Ohio State University
[◇]Translational Data Analytics Institute, The Ohio State University
[♡]Department of Biomedical Informatics, The Ohio State University
{jiang.2880, zhou.3820, peng.707, ning.104, zhu.3440}@osu.edu

## Abstract

Achieving better alignment between vision embeddings and Large Language Models (LLMs) is crucial for enhancing the abilities of Multimodal LLMs (MLLMs), particularly for recent models that rely on powerful pretrained vision encoders and LLMs. A common approach to connect the pretrained vision encoder and LLM is through a projector applied after the vision encoder. However, the projector is often trained to enable the LLM to generate captions, and hence the mechanism by which LLMs understand each vision token remains unclear. In this work, we first investigate the role of the projector in compressing vision embeddings and aligning them with word embeddings. We show that the projector significantly compresses visual information, removing redundant details while preserving essential elements necessary for the LLM to understand visual content. We then examine patch-level alignment—the alignment between each vision patch and its corresponding semantic words—and propose a *multi-semantic alignment hypothesis*. Our analysis indicates that the projector trained by caption loss improves patch-level alignment but only to a limited extent, resulting in weak and coarse alignment. To address this issue, we propose *patch-aligned training* to efficiently enhance patch-level alignment. Our experiments show that patch-aligned training (1) achieves stronger compression capability and improved patch-level alignment, enabling the MLLM to generate higher-quality captions, (2) improves the MLLM's performance by 16% on referring expression grounding tasks, 4% on question-answering tasks, and 3% on modern instruction-following benchmarks when using the same supervised fine-tuning (SFT) setting. The proposed method can be easily extended to other multimodal models.

## 1 Introduction

Multimodal Large Language Models (MLLMs) [1, 2, 3, 4, 5, 6, 7] have recently gained significant attention and made notable progress. These models possess the ability to process and understand both visual and textual information, enabling them to perform complex reasoning [8, 9], generate textual descriptions from images [10], and answer image-related questions [11].

Consider an MLLM $\mathcal{M} = (\mathcal{E}, \mathcal{L}, \mathcal{P})$ where $\mathcal{E}$ is the vision encoder, $\mathcal{L}$ is the LLM, and $\mathcal{P}$ is the lightweight projector that connects the two parts. The standardized training paradigm follows two key phases: pretraining and instruction-tuning [1, 2]. During the pretraining phase, only the lightweight projector $\mathcal{P}$ is trained leaving the vision encoder $\mathcal{E}$ and the LLM $\mathcal{L}$ frozen. In instruction-tuning stage, both projector $\mathcal{P}$ and the LLM $\mathcal{L}$ are trainable. Despite the remarkable progress achieved through

---

[*]The corresponding author.

39th Conference on Neural Information Processing Systems (NeurIPS 2025).

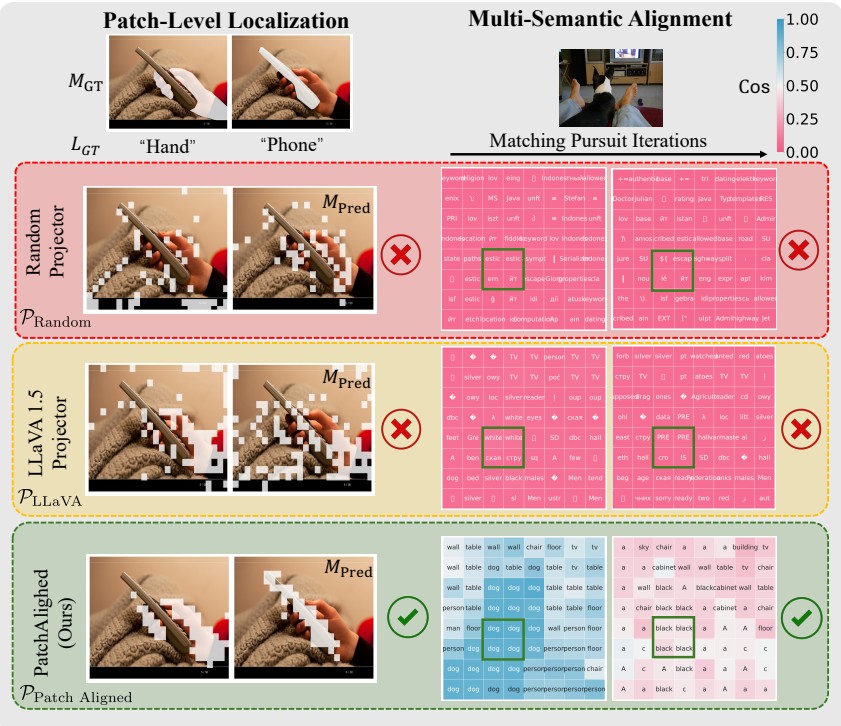

Figure 1: The patch-level alignment is measured in two aspects: Left) **patch-level localization**. Using LLM word embeddings of labels $L_{GT}$, we predict the most relevant image regions by calculating cosine similarity with the vision embedding. Right) **multi-semantic alignment**. Using matching pursuit, we decompose each vision embedding into several discrete words by treating LLM word embeddings as a basis. Since the vision embedding are obtained after the MLLM projector, we compare across three projectors: random projector $\mathcal{P}_{Random}$ (top), LLaVA1.5 projector $\mathcal{P}_{LLaVA}$ (middle), and our patch-aligned projector $\mathcal{P}_{Patch\ Aligned}$ (bottom). Results show that $\mathcal{P}_{LLaVA}$ exhibits weak patch-level alignment abilities, while our $\mathcal{P}_{Patch\ Aligned}$ significantly enhances these two aspects.

this training paradigm, recent works [12, 13, 14, 15, 16] reveal that these MLLMs still struggle with region-specific understanding and tend to hallucinate with irrelevant or incorrect information. The reason of these issues remains an active area. In addition to limitations in the vision encoder or the capabilities of the language model, a significant contributing factor lies in the projector [4, 17].

The platonic representation hypothesis [18] suggests that representations in deep networks are converging across data modalities[2], implying a shared structural alignment. However, for a vision encoder $\mathcal{E}$ and an LLM $\mathcal{L}$ that are pretrained separately, there is no guarantee that the embedding space induced by $\mathcal{E}$ will share the same basis as the embedding matrix $W$ in $\mathcal{L}$, even if they have the same dimensionality. Thus, as the only connection between the two modalities, the projector $\mathcal{P}$ plays a crucial role. However, current work remains at a superficial understanding that the projector performs alignment, lacking a thorough and systematic analysis of its function. Thus, in this paper, we are motivated by the following questions:

*How does the projector align multimodal information in MLLMs?*
*How to quantify and improve alignment in current models?*

**Contributions.** In this work, we provide a detailed analysis of the alignment between each vision patch and its corresponding semantic words and develop methods to improve patch-level alignment, enabling the LLM to better understand visual content in the input space. Our contributions as follows,

**Projector compresses visual information.** Vision embeddings are naturally continuous and contain redundant information, whereas LLM input word embeddings are discrete. Thus, a natural question arises: *Is the information contained in the vision embeddings compressed through the projector?* To address this question, we propose quantifying the amount of information contained in the embeddings

---

[2]In the sense that vision and language models measure the distance between data points in a similar way.

using Von Neumann entropy [19]. Our experiments show that information is significantly compressed after projection, indicating that the projector plays a crucial role in eliminating redundant information while preserving the essential elements needed for the LLM to understand the visual content.

**Analyzing patch-level alignment.** We then examine each image patch in detail and study how its embedding aligns with the corresponding text embedding. However, challenges arise due to (1) a lack of text labels for each image patch and (2) the possibility that each image patch contains multiple semantic meanings. To address these challenges, we propose two complementary approaches to quantitatively and qualitatively study patch-level alignment.

We first focus on alignment with respect to the objects in the image. Specifically, for an input image $X$ with embedding $V = \mathcal{P} \circ \mathcal{E}(X)$, which serves as input to an LLM $\mathcal{L}$, we propose a patch-level alignment measure $\mathrm{Align}(V, W)$ to quantify the alignment between $V$ and the word embedding $W$. A higher $\mathrm{Align}(V, W)$ indicates a greater ability of the LLM to identify objects, even in the word embedding space. Inspired by previous work on decomposing word embedding vectors [20, 21], we then propose a *multi-semantic alignment* hypothesis, which states that the embedding for each vision patch can be decomposed as a linear combination of word embeddings corresponding to all semantic meanings within the patch. To verify this hypothesis, we apply the matching pursuit algorithm [22] to identify the most relevant tokens from the LLM dictionary for each vision patch.

As shown in Figure 1, our analysis indicates that the LLaVA projector improves patch-level alignment but only to a limited extent, resulting in weak and coarse alignment. This is due to (1) the caption loss only implicitly enforces token-level alignment, (2) captions tend to be short and primarily focus on a few prominent regions of interest, often neglecting many other regions. Consequently, numerous visual tokens (e.g., floor and TV cabinet) are often aligned with meaningless or garbled words.

**Patch-Aligned Training for Improving Patch-Level Alignment.** To address this issue, we propose a simple yet effective method called *Patch-Aligned Training* to enhance fine-grained alignment. In addition to the standard image caption loss in the pretraining state, we introduce additional patch loss, similar to $\mathrm{Align}(V, W)$, to capture the alignment between $V$ and the word embedding $W$. Notably, the patch loss relies only on the LLM embedding matrix $W$ and is therefore computationally negligible compared to the caption loss, which requires inference and backpropagation through the LLM. As demonstrated in Figure 1, experiments show that Patch-Aligned training achieves stronger compression capability and improved patch-level alignment, enabling the LLM to generate higher-quality captions. Moreover, under the same SFT setting, the enhanced projector improves the MLLM's performance by 16% on referring expression grounding tasks, 4% on question-answering tasks, and 3% on modern instruction-following benchmarks.

**Patch-Aligned Dataset with Detailed Patch-Level Semantic Labels.** To enable patch-aligned training, we address the lack of patch-level annotated data by introducing an automated data annotation pipeline that sequentially leverages RAM [23], Grounding DINO [24], and SAM [25]. Applying this pipeline to the 558K LLaVA pretraining dataset, we construct the Patch-Aligned Dataset (PAD), which provides extensive and diverse patch-level annotations. To support future research, we publicly release both the annotation pipeline and the resulting dataset.

## 2 Related Works

**Multimodality Large Language Models.** Many MLLMs, such as LLaVA-1.5/1.6 [2, 26], BLIP-2 [4], InstructBLIP [27], MiniGPT-4 [5], Otter [28], and mPLUG-Owl [29], can be viewed as stitched models, formed by connecting a pretrained (and often frozen) vision encoder (such as ViT) to a pretrained LLM through a projector or connector. The projector can be trained using either ($i$) a 1-stage approach, where it is directly trained alongside the fine-tuning LLM during instruction training [30], or ($ii$) a 2-stage approach, where the projector is first pretrained on adapter data before unfreezing the LLM and connector during instruction tuning [2]. The 2-stage approach has been widely adopted since LLaVA and has been shown to be beneficial [31]. However, during the pretraining stage, these models primarily rely on caption loss to achieve coarse alignment between modalities, which tends to lack region-level understanding abilities. Recent efforts, such as GPT4RoI [13], Kosmos-2 [14], and GLaMM [15], have attempted to improve region-specific, fine-grained understanding. However, these approaches often rely on grounding techniques or introduce additional tokens to enhance inference-time capabilities. They focus on improving inference rather than representation analysis of

fine-grained alignment. In contrast, our approach seeks to enhance fine-grained understanding by improving patch-level alignment without requiring additional training or tokens.

**MultiModal Alignment Analysis.** Existing works analyze cross-modal alignment from two perspectives: coarse alignment and fine-grained alignment. Coarse alignment is evaluated using metrics such as AC Score [32], which heavily depends on the CLIP [33] model, and Modality Integration Rate (MIR) [34], a statistic-based measure akin to FID. While these metrics provide insights into pretraining performance, they fail to address fine-grained token-level alignment. For fine-grained alignment, methods like Logit Lens Analysis [35] show that object-specific information is localized to token positions corresponding to image regions but lack proposals for improvement. Other works align coordinate, text, and image modalities through question-answer formats but overlook feature-level understanding [36]. Concurrently, SEA [37] enhances token-level alignment using predefined word lists and contrastive loss, but its reliance on the CLIP model and fixed vocabularies limits accuracy and flexibility. In contrast, as a fine-grained alignment model, our approach employs annotations from the RAM [23] model, which avoids predefined word lists for more accurate tagging, and introduces a cosine similarity loss, offering a simpler and more efficient alternative to contrastive loss.

# 3 Understanding Multimodal Projector by Compression and Alignment

In this section, we study the projector from both macro and micro perspectives: 1) information compression in Section 3.1 and 2) patch-level alignment in Section 3.2.

## 3.1 Macro-scale Analysis: Information Compression

Consider a MLLM $\mathcal{M} = (\mathcal{E}, \mathcal{L}, \mathcal{P})$. For each input images $\boldsymbol{X}$, the vision embedding of the $n$-th image before and after the projector is a sequence of embeddings as

$$\boldsymbol{V}_{\text{before}} = \mathcal{E}(\boldsymbol{X}) \in \mathcal{R}^{d \times S}, \quad \boldsymbol{V}_{\text{after}} = \mathcal{P} \circ \mathcal{E}(\boldsymbol{X}) \in \mathcal{R}^{d' \times S} \tag{1}$$

where $S$ is the number of vision tokens and $d, d'$ are the embedding dimensions of the vision encoder $\mathcal{E}$ and LLM $\mathcal{L}$. For $N$ images, we compute the embeddings for each image and stack them together. We denote the resulting embeddings as $\boldsymbol{V}_{\text{before}} \in \mathcal{R}^{d \times NS}$ and $\boldsymbol{V}_{\text{after}} \in \mathcal{R}^{d' \times NS}$.

We hypothesize that the projector plays a crucial role in eliminating redundant information while preserving essential elements for the LLM to understand vision content. To quantify this, we measure the information using the basis-independent, transformation-invariant Von Neumann entropy [19].

**Definition 3.1** (Von Neumann Entropy of Feature Embeddings)**.** Let $\boldsymbol{V} \in \mathbb{R}^{d \times n}$ be a set of $n$ feature vectors, each of dimension $d$, and let $\boldsymbol{v}_i \in \mathbb{R}^d$ denote the $i$-th column of $\boldsymbol{V}$. Define the normalized empirical covariance matrix by,

$$\boldsymbol{\rho_V} = \frac{\boldsymbol{\Sigma_V}}{\text{Tr}(\boldsymbol{\Sigma_V})}, \quad \boldsymbol{\Sigma_V} = \frac{1}{n} \sum_{i=1}^{n} \boldsymbol{v}_i \boldsymbol{v}_i^\top \in \mathbb{R}^{d \times d},$$

where $\boldsymbol{\Sigma}_V$ is normalized by its trace to obtain the density matrix of trace 1. Then, we compute the information contained in the embeddings $\boldsymbol{V}$ through the *Von Neumann entropy* [19] as follows,

$$\text{H}(\boldsymbol{V}) = -\text{Tr}(\boldsymbol{\rho_V} \log \boldsymbol{\rho_V}) = -\sum_j \lambda_j \log(\lambda_j),$$

where $\{\lambda_j\}$ are the eigenvalues of $\boldsymbol{\rho_V}$.

The Von Neumann Entropy measures how evenly information spreads across the feature space of learned embeddings, indicating their effective dimensionality. Higher values show a well-distributed, high-rank representation with diverse features, while lower values indicate compression—resulting in information loss and a reduced effective rank of the covariance matrix.

To measure the compression abilities of different projectors, we compare pretrained (stage 1) and randomly initialized variants. We evaluate several projector types commonly used in the MLLM field, including Linear, 2-layer MLP, and C-Abstractor [38]. We evaluated these across 100 selected images from the COCO2017 dataset [39]. The Von Neumann Entropy of vision embeddings before and after projection are shown in Table 1.

Table 1: Comparison of Von Neumann Entropy of vision embedding before and after projection.

| Projector | $\mathcal{P}_{\text{LLaVA Linear}}$ | $\mathcal{P}_{\text{Random Linear}}$ | $\mathcal{P}_{\text{LLaVA MLP}}$ | $\mathcal{P}_{\text{Random MLP}}$ | $\mathcal{P}_{\text{LLaVA C-Abs}}$ | $\mathcal{P}_{\text{Random C-Abs}}$ |
|---|---|---|---|---|---|---|
| $\text{H}(\boldsymbol{V}_{\text{before}})$ | 4.8353 | 4.8353 | 4.8353 | 4.8353 | 4.8353 | 4.8353 |
| $\text{H}(\boldsymbol{V}_{\text{after}})$ | 2.4829 | 4.8197 | 2.0362 | 4.8245 | 3.5850 | 7.4913 |

Based on Table 1, we make several key observations:

- **Pretrained v.s. Random.** The vision feature after the pretrained projectors ($\mathcal{P}_{\text{LLaVA Linear}}$, $\mathcal{P}_{\text{LLaVA MLP}}$ and $\mathcal{P}_{\text{LLaVA C-Abstractor}}$) exhibit lower entropy compared to random initialized ones ($\mathcal{P}_{\text{Random Linear}}$, $\mathcal{P}_{\text{Random MLP}}$ and $\mathcal{P}_{\text{Random C-Abstractor}}$), indicating that the pretrained project actively *compresses* the vision features. By contrast, the random projector barely changes the entropy, suggesting no meaningful compression occurs.
- **MLP v.s. Linear.** The vision feature after the MLP projector $\mathcal{P}_{\text{LLaVA MLP}}$ yields a larger drop in entropy than a linear projector $\mathcal{P}_{\text{LLaVA Linear}}$, suggesting that a deeper, non-linear transformation can better remove "redundant" information. A simple linear mapping merely rotates or shifts the embedding space; it has limited capacity that discards irrelevant information. This provides an explanation for the performance advantage of MLP over linear porjector [2].

The compression appears essential for alignment since text embeddings, unlike visual inputs, are compact and discrete—structured around a finite vocabulary and token-based representation. The vision projector must therefore transform high-dimensional, continuous visual data into a format that aligns with text embeddings, naturally producing a more condensed output. However, entropy analysis alone cannot reveal how this alignment occurs at the patch level. Therefore, we examine patch-level alignment from a micro-scale perspective in the next section.

## 3.2 Micro-scale Analysis: Patch-Level Alignment

Unlike the CLIP model, where an image and its corresponding caption are encoded as an embedding vector, alignment can be simply measured using the cosine similarity between the two embedding vectors. Here, however, since the image patch embeddings are given as input to the LLM $\mathcal{L}$, we aim to measure their alignment with word embeddings $\boldsymbol{W}$. This presents several challenges: (1) There is a lack of text labels for each patch; (2) each word may be decomposed into multiple tokens or subwords in the LLM; (3) each image patch may contain multiple semantic meanings. To address these challenges, we propose two complementary approaches to study patch-level alignment.

### 3.2.1 Patch-Level Localization

Given an input image $\boldsymbol{X}$ with $S$ patches, we use $\boldsymbol{V} = \mathcal{P} \circ \mathcal{E}(\boldsymbol{X}) \in \mathbb{R}^{d \times S}$ to denote the $S$ vision embeddings. Due to the lack of labels for each patch, we adopt a simpler approach that relies only on labels for the objects within the image. In particular, suppose the image $\boldsymbol{X}$ contains $P$ objects, each defined by its object tags and bounding box locations. We will develop a mask-label annotation pipeline in the next section. Let the ground-truth mask-label pairs be denoted as $\{(M^{(p)}, L^{(p)})\}_{p=1}^{P}$.

For each label $L^{(p)}$ which may contain multiple words or a single word that is decomposed into multiple subtokens in the LLM, we compute its text embedding $\boldsymbol{t}^{(p)}$ by averaging the LLM word embeddings of all its subtokens:

$$\boldsymbol{t}^{(p)} = \frac{1}{K}\sum_{k=1}^{K}\boldsymbol{w}_k^{(p)}, \quad \{\boldsymbol{w}_k^{(p)}\}_{k=1}^{K} = \boldsymbol{W}(\phi(L^{(p)})), \tag{2}$$

where $\phi$ is the tokenizer that converts $L^{(p)}$ into $K$ subtokens, and $\boldsymbol{w}_k^{(p)} \in \boldsymbol{W}$ represents the $k$-th subtoken embedding. As each object may occupy multiple patches, we identify the relevant patches by computing the cosine similarity $\text{COS}(\boldsymbol{t}^{(p)}, \boldsymbol{v}_i)$ between vision patch $\boldsymbol{v}_i \in \boldsymbol{V}$ and the text embedding $\boldsymbol{t}^{(p)}$. We then select the patches whose similarity score exceeds an adaptive threshold $c > 0$, i.e.,

$$\text{Idx}^{(p)} = \{i \mid \text{COS}(\boldsymbol{t}^{(p)}, \boldsymbol{v}_i) > c, \forall \boldsymbol{v}_i \in \boldsymbol{V}\}, \tag{3}$$

which further gives the predicted bounding box locations $M_{\text{pred}}$. A visualization of the ground truth mask $M$ and predicted mask $M_{\text{pred}}$ is shown in Figure 1 (left).

We now quantify the patch alignment $\text{Align}(\boldsymbol{V}, \boldsymbol{W})$ between the image embeddings $\boldsymbol{V}$ and the word embeddings $\boldsymbol{W}$ through the Intersection over Union (IoU) against the ground-truth mask $M$:

$$\text{Align}(\boldsymbol{V}, \boldsymbol{W}) = \frac{1}{P} \sum_{p=1}^{P} \frac{\text{Intersection}(M_{\text{pred}}^{(p)}, M^{(p)})}{\text{Union}(M_{\text{pred}}^{(p)}, M^{(p)})}. \tag{4}$$

To measure the patch-level alignment of projector, we compare the above measure over three variants: the projector after pretraining ($\mathcal{P}_{\text{LLaVA Stage1}}$), the projector after SFT ($\mathcal{P}_{\text{LLaVA Stage2}}$), and a random MLP projector ($\mathcal{P}_{\text{Random MLP}}$). The results on GranDf dataset [15] are shown in Table 2.

Table 2: Patch alignment of projectors.

| Projector | $\text{Align}(\boldsymbol{V}, \boldsymbol{W})$ |
|---|---|
| $\mathcal{P}_{\text{Random MLP}}$ | 0.065 |
| $\mathcal{P}_{\text{LLaVA Stage1}}$ | 0.142 |
| $\mathcal{P}_{\text{LLaVA Stage2}}$ | **0.152** |

**Projector improves patch-level alignment.** As shown in Table 2, the pretrained projector achieves a higher $\text{Align}(\boldsymbol{V}, \boldsymbol{W})$ than a random one, with the measure further improving after SFT, indicating better alignment between the vision and word embedding spaces. However, we also observe that the LLaVA projector exhibits low mIoU values in both Stage 1 and Stage 2, suggesting that text labels derived from LLM embeddings cannot accurately identify their corresponding image patch positions. *This underscores the limitation of the original LLaVA projector in patch-level alignment.*

### 3.2.2 Multi-Semantic Alignment

While the above approach provides a quantitative method to measure patch-level alignment, the label for each object is often very short. For example, a TV may be labeled simply as "TV," without additional attributes such as color. On the other hand, the continuous vision embedding $v$ is expected to carry multiple semantic meanings that are understandable by the LLM. We express this as the following hypothesis.

**Hypothesis 3.1.** *In an MLLM, the embedding $v$ for each vision patch can be decomposed as a sparse linear combination of word embeddings: $v \approx \sum_{k \in \Omega} \alpha_k w_k$, where $\Omega$ is the set of subtokens representing all semantic meanings within the patch, and $\alpha_k$ is the coefficient for each subtoken.*

This hypothesis is similar to previous ones on (contextualized) word embedding vec-

---

**Algorithm 1** Matching Pursuit for Vision Embedding

**Input:** Vision embedding $\boldsymbol{v} \in \mathbb{R}^d$; LLM word embedding matrix $\boldsymbol{W} = [\boldsymbol{w}_1, \boldsymbol{w}_2, \ldots, \boldsymbol{w}_M] \in \mathbb{R}^{d \times M}$; Number of selected word embeddings $K$.
**Output:** Top-$K$ matched word embeddings $\{\boldsymbol{w}^{(i)}\}_{i=1}^{K}$.

---

Initialize $\boldsymbol{v}^{(1)} \leftarrow \boldsymbol{v}$
Initialize an empty set $\mathcal{S} \leftarrow \emptyset$
**for** $i = 1$ **to** $K$ **do**
    // Find the most relevant word embedding
    $\boldsymbol{w}^{(i)} \leftarrow \arg \max_{\boldsymbol{w} \in \boldsymbol{W}} \langle \boldsymbol{w}, \boldsymbol{v}^{(i)} \rangle$
    // Store selected embedding
    $\mathcal{S} \leftarrow \mathcal{S} \cup \{\boldsymbol{w}^{(i)}\}$
    // Remove projection
    $\boldsymbol{v}^{(i+1)} \leftarrow \boldsymbol{v}^{(i)} - \langle \boldsymbol{w}^{(i)}, \boldsymbol{v}^{(i)} \rangle \boldsymbol{w}^{(i)}$
**end for**
**Return** $\mathcal{S}$

---

tors as a sparse linear combination of word/transformer factors [20, 21], but extended to multimodal embeddings. To support this hypothesis, we utilize the matching pursuit algorithm [22] to identify the top-$K$ most relevant subtokens. Specifically, at the $i$-th iteration, the algorithm selects the discrete word embedding $\boldsymbol{w}^{(i)}$ from the LLM embedding space $\boldsymbol{W}$ that has the highest similarity with the current vision embedding $\boldsymbol{v}^{(i)}$, ensuring it is the most semantically aligned token. Once selected, its contribution is removed from the vision embedding. Through this iterative process, we identify the key semantic components within the vision embedding. We present the details in Algorithm 1.

As in [20, 21], we provide qualitative results due to the lack of ground-truth multi-semantic labels. As shown in Figure 1, while LLaVA vision embeddings can encode basic object information (e.g., "TV") and color attributes (e.g., "white"), many patches remain uninterpretable. This analysis reveals that the projector has limited multi-semantic alignment capabilities. More results in Appendix D.

## 4 Patch-Aligned Training and Analysis

The previous section demonstrates that training with image caption task leads to weak patch-level alignment. In this section, we propose *patch-aligned training* to enhance fine-grained alignment.

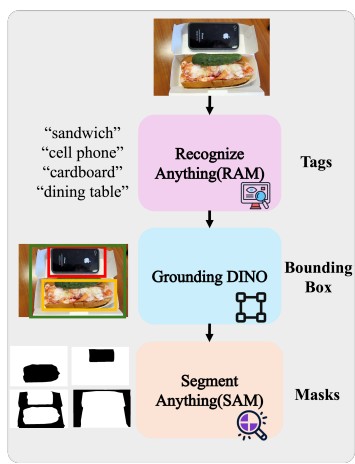

Figure 2: Annotation pipeline.

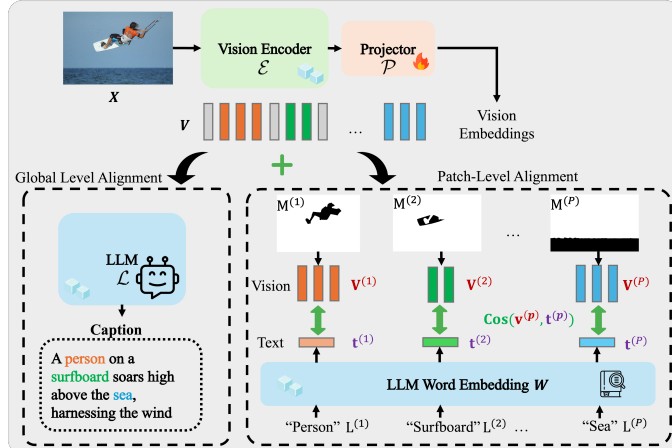

Figure 3: Overview of patch-level alignment method.

## 4.1 Patch-Aligned Training

**Mask-Label Annotation Pipeline.** We develop an automated pipeline to create the Patch-Aligned Dataset (PAD), which enriches the LLaVA-pretrained dataset with fine-grained annotations including object tags, bounding boxes, and segmentation masks. Our pipeline combines state-of-the-art models (RAM[23], Grounding DINO[24], and SAM[25]) for object recognition and segmentation. As shown in Figure 2, RAM first generates object tags, which Grounding DINO uses to create bounding boxes. After filtering with non-maximum suppression, SAM generates segmentation masks for each object. More details about the format of PAD can be found in Appendix A.

**Patch-Aligned Training.** Given an input image $X$ with $P$ associated mask-label pairs $\{(M^{(p)}, L^{(p)})\}_{p=1}^{P}$, the vision embedding after projection is represented as $V = \mathcal{P} \circ \mathcal{E}(X) \in \mathbb{R}^{d \times S}$. For each mask $M^{(p)}$, let $\text{Idx}^{(p)}$ represent the set of vision tokens that are covered by the mask for at least half of the patch area. We use $V^{(p)}$ to denote the embeddings of the selected vision tokens in Figure 3. Using the same approach as in eq. (2) to compute the text embedding $t^{(p)}$ for the label $L^{(p)}$, we similarly represent the vision embedding of the object by taking the mean of the selected vision embeddings $v^{(p)} = \frac{1}{L^{(p)}} \sum_{i \in \text{Idx}^{(p)}} v_i \in \mathbb{R}^d$. We then introduce the *patch-alignment loss* to maximize the cosine similarity between the mask-selected vision embedding $v^{(p)}$ and the corresponding text embedding $t^{(p)}$:

$$L_{\text{patch}} = 1 - \frac{1}{P} \sum_{n=1}^{P} \text{COS}(v^{(p)}, t^{(p)}). \tag{5}$$

To achieve global-level alignment, we retain the commonly used *caption loss*. Specifically, given tokenized caption tokens $(x_1, x_2, ..., x_T)$ for the input image, the caption loss computes the ability to predict each subsequent word in the caption sequence

$$L_{\text{caption}} = - \sum_{t=1}^{T} \log p_{\mathcal{L}}\left(x_t \mid V, x_{<t}\right), \tag{6}$$

where $p_{\mathcal{L}}\left(x_t \mid V, x_{<t}\right)$ denotes the predicted probability for the $t$-th token $x_t$ based on the previous tokens $x_{<t}$ and the vision embedding $V$.

Our patch-aligned training combines both the caption loss and the patch-alignment loss

$$L = L_{\text{caption}} + \beta L_{\text{patch}}, \tag{7}$$

where $\beta > 0$ is used to balance the global-level alignment and patch-level alignment.

**Efficiency of the patch-alignment loss.** The patch-alignment loss $L_{\text{patch}}$ is computationally more efficient than the caption loss, as it does not rely on the LLM. It only requires calculating cosine similarity with the LLM word embedding matrix $W$, making it lightweight to compute and optimize.

## 4.2 Ablation Study on $\beta$

In this section, we present our ablation studies on hyperparameter $\beta$ in Equation (7) in two aspects:

**Linear increasing vs. fixed schedule.** The linear schedule gradually increases $\beta$ to impose patch-level alignment progressively. This design choice stabilizes early training and prevents premature over-compression. As shown in Table 3, when comparing fixed $\beta = 5$ versus linearly increasing $\beta$ from 0 to 5, the linear schedule showed better performance. Therefore, we adopted the linear increasing schedule for all subsequent experiments.

**Optimal final $\beta$ value:** Since $\beta$ in the objective function balances global-level and patch-level alignment, a larger $\beta$ emphasizes patch-alignment loss over caption loss. We examined the impact of $\beta \in \{0, 2, 5, 10\}$, where the baseline LLaVA-1.5 corresponds to $\beta = 0$. As shown in Table 3, when $\beta$ is too small ($\beta = 0$ or 2), patch-level alignment remains insufficient, leading to suboptimal performance. Conversely, when $\beta$ is too large ($\beta = 10$), the model overly focuses on local regions, compromising its ability to establish comprehensive correspondence between the entire image and sentences. This imbalance degrades global-level alignment and ultimately harms overall performance.

Table 3: Comparison of different $\beta$ schedules on various benchmarks.

| Setting | GQA | Science QA | VizWiz VQA | OKVQA | Avg |
|---|---|---|---|---|---|
| fixed ($\beta = 0$) | 61.93 | 66.80 | 50.00 | 53.42 | 58.04 |
| fixed ($\beta = 5$) | 62.16 | 68.07 | **54.86** | 56.51 | 60.40 |
| linear increasing ($\beta \in [0, 2]$) | 62.64 | 68.12 | 50.84 | 57.52 | 59.78 |
| linear increasing ($\beta \in [0, 10]$) | 62.64 | 68.57 | 50.82 | 56.85 | 59.72 |
| **linear increasing ($\beta \in [0, 5]$)** | **62.99** | **68.67** | 52.29 | **58.29** | **60.56** |

## 4.3 Compression–Information Loss Tradeoff

There exists a tradeoff between redundancy removal and semantic information loss when the compression level continues to increase. Within a proper compression range, more compression improves performance by removing redundant information and enhancing alignment. However, over-compression may potentially cause useful semantic information loss and performance degradation. To empirically validate this tradeoff, we conducted a controlled ablation where we varied the patch loss weight $\beta$ in Equation (7). Larger $\beta$ encourages stronger patch-level alignment and induces more compression. As shown in Figure 4, as $\Delta$H increases, the overall performance first improves then declines. Before the tipping point, redundant information is removed, which improves performance compared to the original LLaVA with $\beta = 0$. However, after the tipping point, performance drops rapidly as over-compression causes semantic information loss.

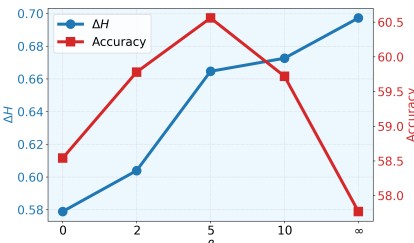

Figure 4: Tradeoff between compression and information loss.

Here, the change of entropy is measured as a normalized one $\Delta H(\boldsymbol{V}) = (H_{\text{before}}(\boldsymbol{V}) - H_{\text{after}}(\boldsymbol{V}))/H_{\text{before}}(\boldsymbol{V}) \in [0, 1]$. Task performance is measured by taking the average of the performance over the QA datasets (GQA, Science QA, VizWiz VQA, and OKVQA).

## 5  Experiments

In this section, we first introduce the experiment setup and training details for Patch-Aligned Training, which is used only in the pretraining stage for training the projector $\mathcal{P}$. We evaluate the effectiveness of our methods in two stages: pretraining stage and SFT stage. In the pretraining stage, we verify that the patch-aligned training achieves better compression and patch-level alignment abilities, enabling

Table 4: Compression and alignment.

| Projector | $\Delta\mathrm{H}(\boldsymbol{V})$ | $\mathrm{Align}(\boldsymbol{V}, \boldsymbol{W})$ | Cos Sim |
|---|---|---|---|
| $\mathcal{P}_{\text{Random}}$ | 0.0108 | 0.065 | 0.06 |
| $\mathcal{P}_{\text{LLaVA}}$ | 2.7991 | 0.142 | 0.07 |
| $\mathcal{P}_{\textbf{Patch Aligned}}$ | **3.8352** | **0.279** | **0.56** |

Table 5: LLaVA v.s. Patch aligned method for caption generation qualities.

| Model | METEOR | ROUGE_L | SPICE |
|---|---|---|---|
| $\mathcal{M}_{\text{LLaVA}}$ | 0.1220 | 0.1661 | 0.1571 |
| $\mathcal{M}_{\text{Patch Aligned}}$ | **0.1256** | **0.1759** | **0.1710** |

Table 6: Comparison on refer expression comprehension benchmarks.

| Models | RefCOCO | | | RefCOCO+ | | | RefCOCOg | |
|---|---|---|---|---|---|---|---|---|
| | val | test-A | test-B | val | test-A | test-B | val | test |
| LLaVA 1.5-7B | 56.22 | 64.43 | 47.38 | 50.00 | 59.2 | 39.0 | 48.8 | 48.4 |
| **Patch Aligned (Ours)** | **65.97** | **72.26** | **55.82** | **58.49** | **66.87** | **48.09** | **55.78** | **56.24** |

the LLM to generate higher-quality captions. In the SFT stage, we verify that fine-tuning with the new projector yields better performance across three aspects: (1) refer expression comprehension, (2) visual question answering and (3) instruction following benchmarks.

## 5.1 Experiment Setup

For a fair comparison, we follow LLaVA-1.5 [1]'s architecture, training setup, and datasets. Our approach differs in two key aspects: (1) we introduce a patch-aligned loss where $\beta$ increases linearly from 0 to 5 during stage 1, and (2) we use the PAD dataset with detailed annotations to pretrain the projector. See Appendix B for details.

## 5.2 Stage1: Pretrained Model Evaluation

### 5.2.1 Compression and Patch-Level Alignment

We evaluate the patch-aligned projector $\mathcal{P}_{\text{Patch Aligned}}$ using: Von Neumann entropy reduction $\Delta\mathrm{H}(\boldsymbol{V}) = \mathrm{H}(\boldsymbol{V}_{\text{before}}) - \mathrm{H}(\boldsymbol{V}_{\text{after}})$, patch alignment $\mathrm{Align}(\boldsymbol{V}, \boldsymbol{W})$ (Section 3.2.1), and vision-text embedding cosine similarity (Section 3.2.2). As shown in Table 4, compared to baselines $\mathcal{P}_{\text{Random}}$ and $\mathcal{P}_{\text{LLaVA}}$ on 100 COCO 2017 images [39], our projector achieves higher entropy reduction and better performance on both mIoU and cosine similarity, demonstrating superior patch-level alignment.

### 5.2.2 Measuring Caption Quality

To examine the advantages of improved patch-level alignment, we first evaluate the stage1 MLLM $\mathcal{M}_{\text{Patch Aligned}} = (\mathcal{E}, \mathcal{L}, \mathcal{P}_{\text{Patch Aligned}})$ directly on caption generation while keeping both the vision encoder and LLM frozen. To measure the caption quality, we utilize three metrics: METEOR, ROUGE-L, and SPICE. As shown in Table 5, $\mathcal{M}_{\text{Patch Aligned}}$ generate higher quality of captions that benefits from the explicit patch-level alignment process in the pretraining stage.

## 5.3 Stage2: SFT Model Evaluation

### 5.3.1 Refer Expression Comprehension

To better demonstrate the method's enhanced fine-grained image understanding and localization capabilities, we further evaluate our approach on the refer expression comprehension(REC) tasks, including the RefCOCO[40], RefCOCO+[41], and RefCOCOg[41]. Specifically, the REC task requires the model to localize the target object under the guidance of a description. Here we report Acc@0.5(higher is better). As shown in Table 6, our method achieves **significant** improvements across all test splits, with approximately a 16% improvement on average. Notably, our approach uses the same architecture and training data as LLaVA-1.5, only adding patch alignment during pretraining. This minimal change yields substantial improvements in grounding and localization abilities.

### 5.3.2 Visual Question Answering

Visual understanding plays an important role in many real-world applications. We test how well our models perform on text-based visual question answering tasks using multiple benchmark datasets. As shown in Table 7, our method outperforms the LLaVA-1.5 baseline under identical conditions, demonstrating that initializing the MLLM with improved patch-level aligned vision embeddings leads to better fine-grained understanding and enhanced overall performance.

Table 7: Comparison on visual question answering benchmarks.

| Models | LM | Img Sz | GQA | SciQA | VizWiz | OKVQA |
|---|---|---|---|---|---|---|
| BLIP-2 [4] | Vicuna-13B | 224 | 41.0 | 61.0 | 19.6 | - |
| InstructBLIP [27] | Vicuna-7B | 224 | 49.2 | 60.5 | 34.5 | - |
| InstructBLIP [27] | Vicuna-13B | 224 | 49.5 | 63.1 | 33.4 | - |
| Shikra [42] | Vicuna-13B | 224 | - | - | - | - |
| IDEFICS-9B [43] | LLaMA-7B | 224 | 38.4 | - | 35.5 | - |
| IDEFICS-80B [43] | LLaMA-65B | 224 | 45.2 | - | 36.0 | - |
| Qwen-VL [44] | Qwen-7B | 448 | 59.3 | 67.1 | 35.2 | - |
| Qwen-VL-Chat [44] | Qwen-7B | 448 | 57.5 | 68.2 | 38.9 | - |
| LLaVA [1] | Vicuna-7B | 224 | - | - | - | - |
| LLaVA-1.5 [2] | Vicuna-7B | 336 | 62.0 | 66.8 | 50.0 | 53.4 |
| **PatchAligned (Ours)** | Vicuna-7B | 336 | **63.0** | **68.7** | **52.3** | **58.3** |

Table 8: Comparison on instruction following benchmarks.

| Models | MMMU | MMVet | CMMMU | MMB$^{EN}$ | MME$^C$ | MME$^P$ |
|---|---|---|---|---|---|---|
| LLaVA 1.5 | 35.30 | 30.70 | 21.80 | **64.00** | 316.10 | 1510.75 |
| **Patch Aligned (Ours)** | **36.56** | **31.61** | **22.70** | 63.14 | **339.64** | **1531.33** |

Table 9: Analysis of compatibility when switching between base LLMs and projector types.

| Models | RefCOCO | | | RefCOCO+ | | | RefCOCOg | |
|---|---|---|---|---|---|---|---|---|
| | val | test-A | test-B | val | test-A | test-B | val | test |
| *Applying to different LLMs (Vicuna 7B [45] → Llama 3.1 8B[46])* | | | | | | | | |
| LLaVA (LLaMA3.1 8B) | 66.32 | 74.30 | 56.09 | 58.92 | 68.62 | 48.21 | 56.90 | 55.87 |
| **Patch Aligned (LLaMA3.1 8B)** | **68.27** | **74.99** | **58.49** | **61.25** | **68.88** | **50.93** | **58.41** | **57.58** |
| *Applying to different projectors (MLP → C-Abstractor [38])* | | | | | | | | |
| LLaVA (C-Abstractor) | 58.08 | 65.83 | 49.44 | 50.28 | 59.10 | 41.01 | 49.38 | 49.81 |
| **Patch Aligned (C-Abstractor)** | **60.89** | **67.70** | **51.21** | **53.56** | **60.74** | **42.01** | **51.91** | **51.61** |

### 5.3.3 Instruction Following Benchmarks

In addition to conventional vision-language evaluations, we assess our method's real-world capabilities by conducting evaluations on modern instruction-following benchmarks. As shown in Table 8, our model demonstrates superior performance in understanding when following user instructions.

### 5.3.4 Compatibility of Patch Aligned Training

We demonstrate our method's general effectiveness by replacing both the MLLM (from Vicuna 7B[45] to Llama 3.1 8B[46]) and the projector (from MLP to C-Abstractor[38]). Using C-Abstractor, we set the number of output visual tokens to 256. All models follow the same training as LLaVA 1.5. We focus our evaluation on referring expression comprehension capabilities. As shown in Table 9, patch-aligned methods achieve consistent improvements compared to their variants.

## 6 Conclusion

In this paper, we examine the projector's role at macro and micro scales, showing it compresses visual information while improving patch-level alignment. Though alignment remains coarse, our proposed patch-aligned training enhances both compression and alignment capabilities. This leads to better caption generation and grounding performance, providing insights into multimodal reasoning.

However, despite the improvements, certain limitations remain to be addressed. First, while we introduce the multi-semantic alignment hypothesis, finding an optimal representation for each visual token in the embedding space remains a significant challenge. Simply aligning with the same averaged word embedding may limit the interpretability and expressive power of LLMs. Moreover, due to the inherent compactness of language, manually guiding the projector for semantic alignment raises concerns about potential information loss in visual tokens. Addressing these challenges requires the development of more effective alignment strategies, which will be crucial for further enhancing the capabilities and robustness of multimodal LLMs.

## Acknowledgements

We acknowledge support from NSF grants IIS-2312840 and IIS-2402952, as well as the ORAU Ralph E. Powe Junior Faculty Enhancement Award.

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

# Appendix

The appendix is organized as follows. First, we show the details of mask-label annotation pipeline and the the patch-aligned dataset format in Appendix A. Then, we introduce the details about experiment setting in Appendix B. Finally, we present visualizations related to token-level alignment—specifically patch-level localization in Appendix C and multi-semantic alignment in Appendix D.

## A   Patch Aligned Dataset

In this section, we present the format of the Patch Aligned Dataset(PAD) in comparison to the LLaVA pretraining dataset following the mask-label annotation pipeline.

**Mask-Label Annotation Pipeline.** To address the lack of patch-level annotated data, we develop an automated annotation pipeline for generating the Patch-Aligned Dataset (PAD), designed to refine the LLaVA-pretrained dataset by incorporating fine-grained details. PAD enriches this dataset with detailed annotations, including object tags, bounding box locations, and segmentation masks for individual objects. By incorporating dense, pixel-level grounding information, PAD is designed to enhance fine-grained image-text alignment during the pretraining stage, thereby improving the model's ability to understand localized regions within the image.

As illustrated in Figure 2, our automated annotation pipeline consists of diverse state-of-the-art models, including Recognize Anything Model (RAM) [23], Grounding DINO [24], and Segment Anything Model (SAM) [25]—to perform grounded image segmentation and object recognition. First, RAM generates object tags from the input image. These tags are then passed to Grounding DINO, which generates bounding boxes for each identified object. Afterward, a Non-Maximum Suppression (NMS) process is applied to filter overlapping bounding boxes based on Intersection over Union (IoU) thresholds. The remaining bounding boxes are passed to SAM, which generates segmentation masks for each object. The pipeline outputs the segmented image with bounding boxes, along with metadata in JSON format, including object tags, bounding box coordinates, and RLE-encoded masks for further analysis. As shown in Table 10. we annotate the images of the LLaVA pretraining dataset with additional object tags, bounding box coordinates, and RLE-encoded masks stored in JSON file. The RLE-encoded masks can be decoded back into binary masks that have the image size.

Table 10: Comparison of LLaVA Pretraining Dataset and Patch Aligned Dataset (Ours).

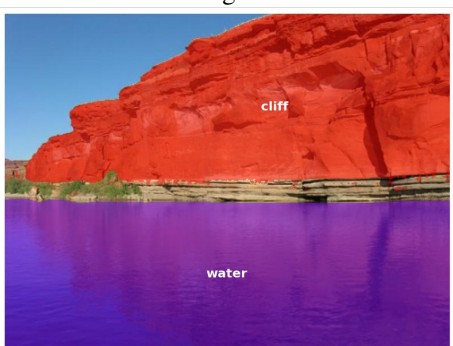

| | |
|---|---|
| **LLaVA Pretraining Dataset** | `"image_id" :  "00000/00000030.jpg"`
`"size":  [448, 336]`
`"caption":  "a canyon wall reflects the water on a sunny day in utah."` |
| **Patch Aligned Dataset (ours)** | `"image_id" :  "00000/00000030.jpg"`
`"size":  [448, 336]`
`"caption":  "a canyon wall reflects the water on a sunny day in utah."`
`"labels": [`
`    {`
`      "tag":  "water",`
`      "bbox":  [-0.0003204345703125, 182.57894897460938,`
`            447.99951171875, 335.67926025390625],`
`      "rle_mask":  "k5d4L50000001000000010001000000000000000..."`
`    },`
`    {`
`      "tag":  "cliff",`
`      "bbox":  [-0.064117431640625, 0.34404754638671875,`
`            447.9346005859375, 182.572509765625],`
`      "rle_mask":  "]S32.:0eE0V:5004LXY2:[fM302M20200N2N3N1010101N20101N20..."`
`    }]` |

To find the optimal hyperparameters in our mask-label annotation pipeline, we conducted a thorough ablation study using the coco-val 2017 [39] dataset, which provides ground truth bounding boxes. We focused on two key hyperparameters:

- Score threshold: Only boxes with confidence scores above this threshold are selected.
- NMS threshold: During non-maximum suppression (NMS), this determines the maximum allowed overlap between two boxes—if their IoU exceeds this threshold, the box with the lower confidence score is removed.

We evaluated performance using F1 score at IoU of 0.5, which classifies predictions as true or false positives based on an IoU threshold of 0.5. First, we tested various score threshold values:

Table 11: Effect of varying score thresholds.

| Score threshold | 0.1 | 0.2 | 0.3 | 0.4 | 0.5 |
|---|---|---|---|---|---|
| F1 @ [IoU=0.5] | 0.2762 | 0.4931 | 0.6326 | **0.6677** | 0.6207 |

Next, we fixed the score threshold at 0.4 and evaluated various NMS threshold values:

Table 12: Effect of varying NMS thresholds.

| NMS threshold | 0.3 | 0.5 | 0.7 | 0.8 | 0.9 |
|---|---|---|---|---|---|
| F1 @ [IoU=0.5] | 0.6530 | 0.6677 | 0.6702 | **0.6722** | 0.6717 |

Based on this analysis, we selected the optimal hyperparameters (Score threshold = 0.4 and NMS threshold = 0.8) for our final implementation.

To evaluate our pipeline with optimal hyper-parameters, we compare it with the original Grounding DINO[24] on coco-val 2017 [39]. The results are as follows:

Table 13: Comparison of between our pipeline and original Grounding DINO.

| | AP@[IoU=0.50:0.95] | AP@[IoU=0.50] | AP@[IoU=0.75] |
|---|---|---|---|
| Original Grounding DINO | 0.485 | 0.644 | 0.529 |
| **Our pipeline** | **0.531** | **0.676** | **0.572** |

where AP@[IoU=0.50:0.95] is the mean precision across IoU thresholds from 0.50 to 0.95 (stepped by 0.05). A prediction is considered correct if its overlap with ground-truth exceeds the IoU threshold. As the results demonstrate, our pipeline consistently outperforms the baseline.

## B Experiment Setup

- **Architecture** To evaluate the effectiveness of our method, we ensure a fair comparison by following the same architecture as LLaVA 1.5. Specifically, we use CLIP-ViT-L@336px [33] as the vision encoder $\mathcal{E}$, Vicuna-1.5-7B[47] as the LLM $\mathcal{L}$, and a 2-layer MLP as the projector $\mathcal{P}$. The parameter $\beta$ follows a linear schedule, increasing from 0 to 5.
- **Training Details** Following the standard training paradigm in LLaVA [1], our training pipeline consists of two stages. In stage 1, keeping the vision encoder $\mathcal{E}$ and LLM $\mathcal{L}$ frozen, we train only the projector $\mathcal{P}$ using our proposed *Patch Aligned Training* method to obtain the patch-aligned projector $\mathcal{P}_{\text{Patch Aligned}}$. In stage 2, we perform supervised fine-tuning on both the LLM $\mathcal{L}$ and the patch-aligned projector $\mathcal{P}_{\text{Patch Aligned}}$. Following LLaVA's hyperparameters, we optimize all models for 1 epoch using the AdamW optimizer with a cosine learning schedule. The learning rates are set to 1e-3 for pretraining and 2e-5 for instruction tuning. Pretraining requires approximately 8 hours using 8×A5000 GPUs (24G), while visual instruction tuning takes about 10 hours for LLaVA-v1.5-7B on 8xH100 (80G).
- **Dataset** For pretraining dataset, utilizing our automated annotation pipeline, we annotate the 558K subset of the LAION-CC-SBU dataset, which is used as the pretraining dataset of LLaVA. The

resulting dataset comprises 2.3M regions, each associated with a segmentation mask, and includes 33.5K unique tags. For fair comparision, we use the same vision instruction tuning dataset as the one in the LLaVA-1.5, containing LLaVA-Instruct [1], TextVQA [48] , GQA [49], OCR-VQA [50], and Visual Genome[51].

## C    Patch-Level Localization: More Visualizations

Following the micro-scale analysis on patch-level localization in Section 3.2.1, we provide more examples comparing the ground truth mask $M_{\text{GT}}$ and predicted mask $M_{\text{pred}}$ generated by three projectors: the random projector $\mathcal{P}_{\text{Random}}$, pretrained LLaVA 1.5 projector $\mathcal{P}_{\text{LLaVA}}$, and our PatchAligned Projector $\mathcal{P}_{\text{Patch Aligned}}$. As shown in Figure 5, the $\mathcal{P}_{\text{Patch Aligned}}$ predicts more accurate masks.

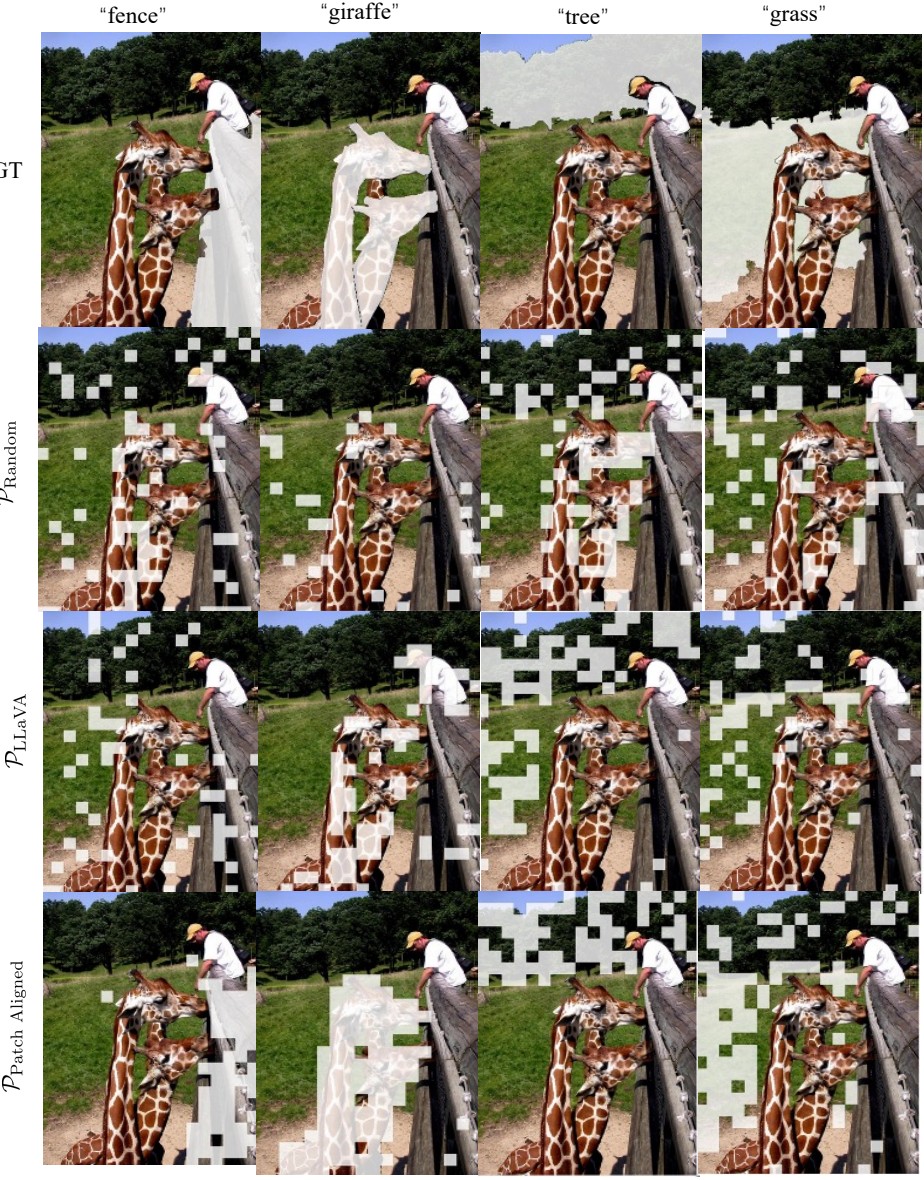

Figure 5: Additional visualization for patch-level localization.

## D   Multi-Semantic Alignment: More Visualizations

We begin by showing the first iteration of matching pursuit, which finds the token in the LLM embedding space that has the highest similarity with the vision embedding. We show the full tokenmap in Figure 6, displaying the found token for each vision patch. We use font size to represent similarity. Tokens recognizable by NLTK are shown in color, while unrecognized tokens remain black. For LLaVA, only partial areas or objects achieve alignment. In contrast, PatchAligned LLaVA achieves better alignment across most patches.

LLaVA              PatchAligned LLaVA

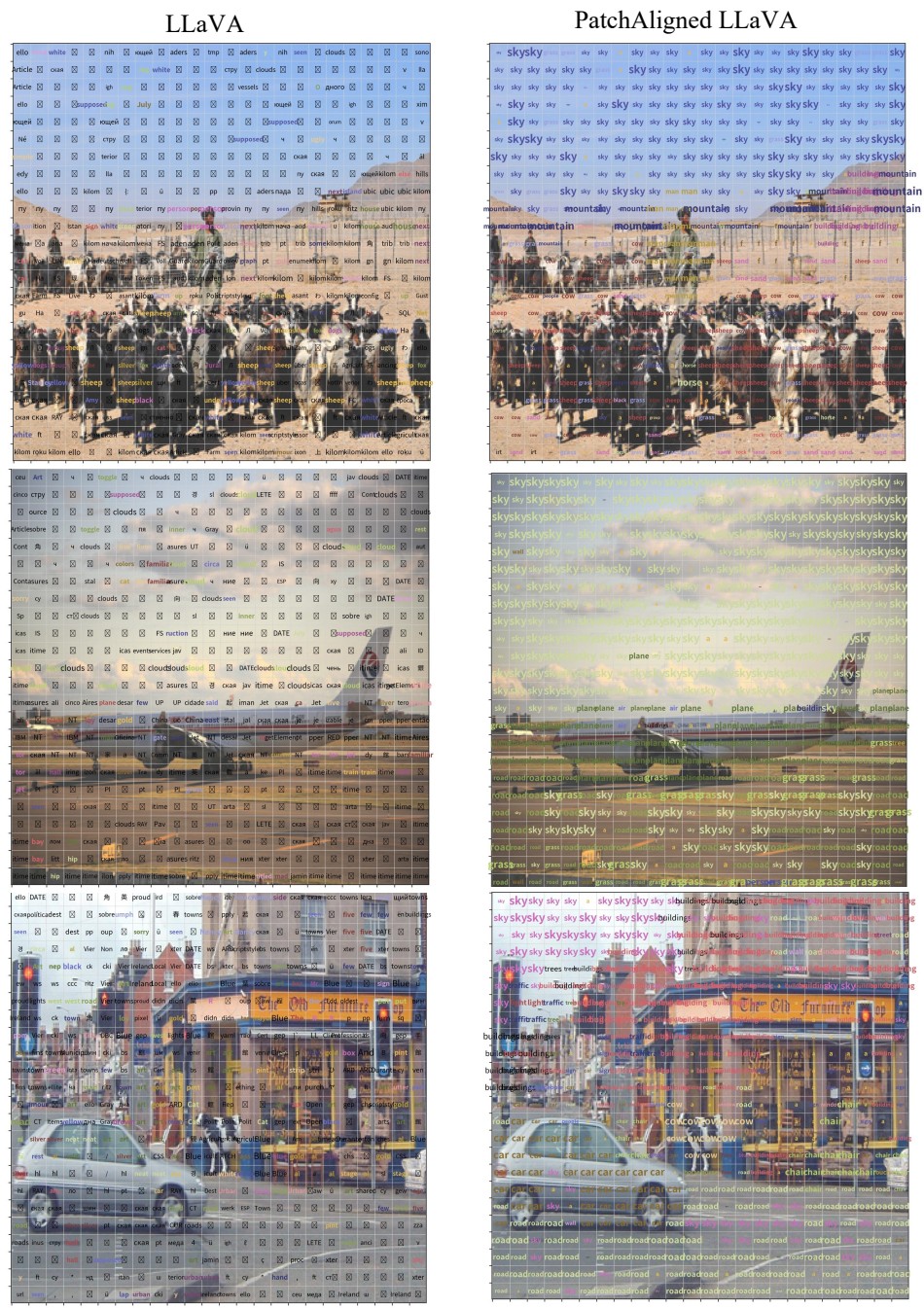

Figure 6: Additional visualization for tokenmap comparing LLaVA and PatchAligned LLaVA.

Next, following Section 3.2.2, we apply matching pursuit on PatchAligned LLaVA for 5 iterations. As shown in Figure 7, the semantic meanings are decoded for each iteration, with cosine similarity decreasing across iterations.

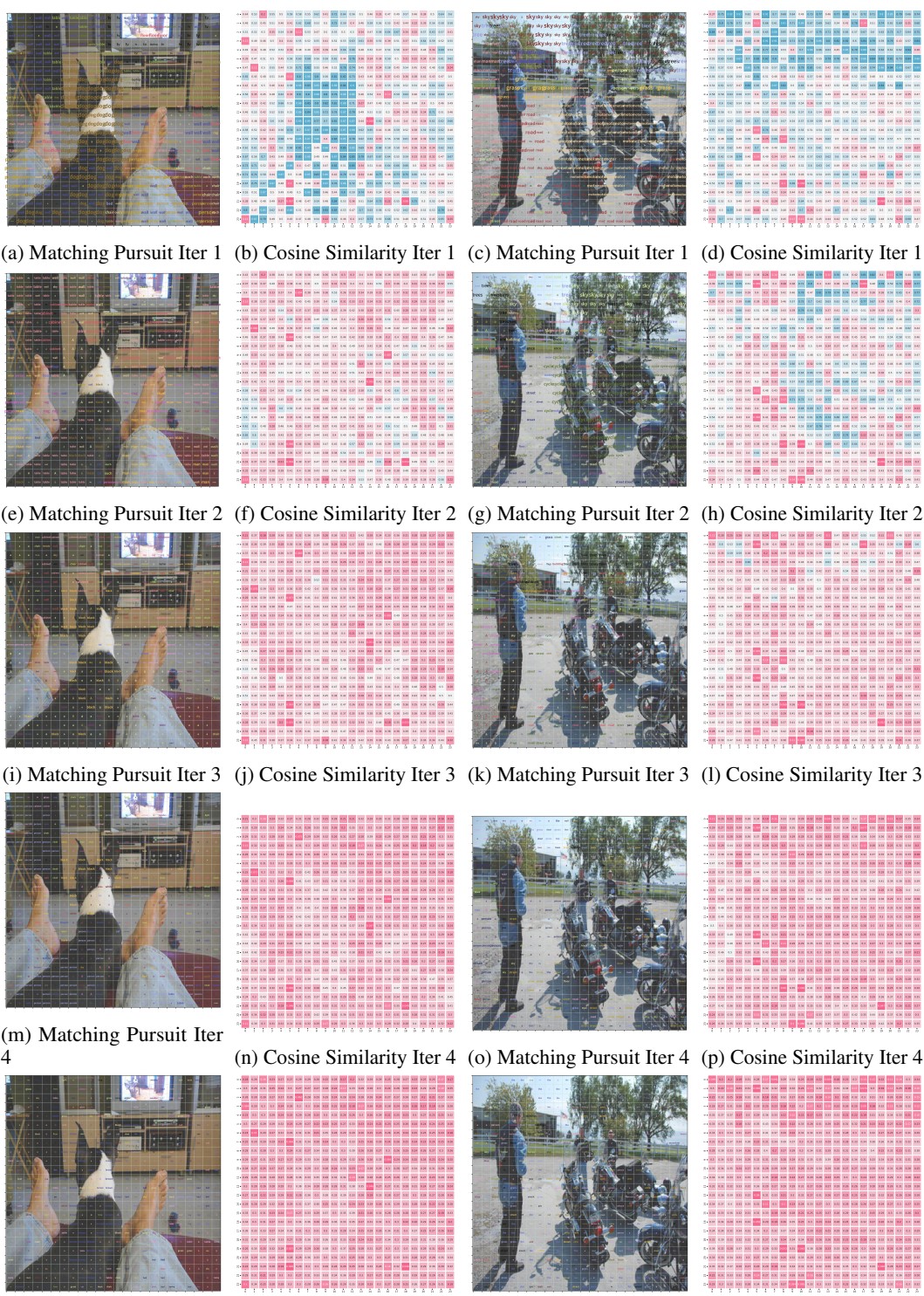

(a) Matching Pursuit Iter 1 (b) Cosine Similarity Iter 1 (c) Matching Pursuit Iter 1 (d) Cosine Similarity Iter 1

(e) Matching Pursuit Iter 2 (f) Cosine Similarity Iter 2 (g) Matching Pursuit Iter 2 (h) Cosine Similarity Iter 2

(i) Matching Pursuit Iter 3 (j) Cosine Similarity Iter 3 (k) Matching Pursuit Iter 3 (l) Cosine Similarity Iter 3

(m) Matching Pursuit Iter 4 (n) Cosine Similarity Iter 4 (o) Matching Pursuit Iter 4 (p) Cosine Similarity Iter 4

(q) Matching Pursuit Iter 5 (r) Cosine Similarity Iter 5 (s) Matching Pursuit Iter 5 (t) Cosine Similarity Iter 5

Figure 7: Perform Matching Pursuit using PatchAligned LLaVA. Each row represents an iteration, with selected tokenmap (Column 1,3) and cosine similarity maps (Column 2,4).

