# OpenReview forum: "Analyzing Fine-Grained Alignment and Enhancing Vision Understanding in Multimodal Language Models"
_NeurIPS.cc/2025/Conference — NeurIPS 2025 poster_

### Official Review · Reviewer_7Lxw · 2025-07-02

**Clarity:** 4
**Significance:** 4
**Originality:** 3
**Rating:** 5
**Confidence:** 4

**Summary:**

This work first analyses the alignment between visual and LLM feature spaces by evaluating several custom metrics before and after projectors in MLLMs. The results reveal that existing MLLMs, such as LLaVA, achieve only weak and coarse alignment, lacking patch-level alignment after pretraining. To address this issue, we introduce a Patch-Aligned Dataset and design a patch-alignment loss to enhance fine-grained alignment. Experiments demonstrate that the proposed patch-alignment training not only improves fine-grained alignment but also boosts the MLLM’s performance after instruction tuning.

**Questions:**

See weakness section.

**Ethical Concerns:**

["NO or VERY MINOR ethics concerns only"]

**Final Justification:**

I have read author's rebuttal other reviewer's comments. Most of my concerns have been addressed and I decide to keep my original score.

**Limitations:**

Limitations already discussed.

**Paper Formatting Concerns:**

No formatting issues.

**Quality:**

3

**Strengths And Weaknesses:**

Strengths:
1)Visual-LLM feature alignment is an underexplored topic in MLLMs. This work provides a quantitative analysis of both coarse and fine-grained alignment using custom metrics.
2)The proposed patch-aligned training is simple yet effective, as validated by experiments.
3)The method improves not only fine-grained alignment but also downstream MLLM performance after instruction tuning.

Weaknesses:
1)The idea of enforcing token-level alignment between visual features and text embeddings is quite similar to SEA, even thought the implantation is different.
2)Since the patch-aligned training uses additional labeled data, the performance gains after SFT—especially on fine-grained recognition tasks—are somewhat expected. A deeper discussion of this limitation is needed.

---

> ### Author Rebuttal · Authors · 2025-07-30
>
> We sincerely thank the reviewer for recognizing the novelty of our work, the effectiveness of our proposed patch-aligned training, and its impact on both alignment and downstream MLLM performance. Below, we respond to the remaining concerns individually and clarify the corresponding points.
>
> > Q1: The idea of enforcing token-level alignment between visual features and text embeddings is quite similar to SEA, even though the implantation is different.
>
> A1: We thank the reviewer for pointing out the related work. While SEA shares the motivation of improving alignment between visual tokens and language embeddings beyond caption-level supervision, our work differs in several significant ways:
>
> 1. Fine-grained alignment measurement: SEA focuses primarily on introducing a new training loss, whereas our work provides a systematic analysis of existing projectors using novel metrics, including *Von Neumann Entropy* (quantifying information compression) and $\mathrm{Align}(\mathbf{V}, \mathbf{W})$ (measuring fine-grained alignment). These analyses reveal the structural limitations of current projectors and provide actionable insights.
>
> 2. Multi-semantic alignment: SEA treats each patch as corresponding to a single label. In contrast, we introduce a Matching Pursuit-based decomposition to extract multiple aligned tokens per patch, capturing fine-grained multi-token semantics—crucial for complex understanding.
>
> 3. Vocabulary coverage: SEA relies on a fixed, predefined vocabulary, which inherently limits semantic granularity. We use RAM to recognize a broad and dynamic vocabulary of visual concepts, allowing our patch supervision to adapt to diverse real-world entities and contexts.
>
> 4. Loss formulation and performance: While SEA adopts a contrastive loss with handcrafted negative sampling, we use a simpler cosine loss that eliminates the need for sampling strategies. Since SEA doesn’t release their code, we implement the contrastive loss on our own. Our ablation studies show that our approach outperforms contrastive loss in downstream tasks:
> | Dataset     | Original LLaVA | SEA (contrastive loss) | Ours (cos loss) |
> |-------------|----------------|-------------------------|------------------|
> | GQA         | 61.93          | 60.95                   | **62.99**        |
> | Science QA  | 66.8           | 67.58                   | **68.67**        |
> | OKVQA       | 53.42          | 51.59                   | **58.29**        |
>
> We will incorporate these key distinctions and comparisons with SEA in the revision.
>
> > Q2: Since the patch-aligned training uses additional labeled data, the performance gains after SFT—especially on fine-grained recognition tasks—are somewhat expected. A deeper discussion of this limitation is needed.
>
> A2: Thank you for your thoughtful comment. We agree with the reviewer that our method introduces additional supervision through bounding boxes. However, the core contribution of our work isn't adding strong supervision, but rather designing a lightweight mechanism to improve fine-grained visual–language alignment in multimodal large language models (MLLMs).
>
> Our performance gains don't come from new module design or additional training data, but from addressing the misalignment in current MLLMs trained solely with caption-level objectives. Compared to the baseline LLaVA model, our method:
>
> - maintains exactly the same architecture,
> - uses the same pretraining dataset, with labels generated by our automatic detection pipeline (no additional datasets or human-annotated labels),
> - adds only a minimal patch-level cosine loss to guide alignment.
>
> This demonstrates that our approach doesn't rely on rich supervisory information, but rather serves as a method that directly addresses a fundamental alignment gap in the existing training pipeline.
>
> Importantly, the supervision signal is automatic, scalable, and practical. It requires no human annotation. Our pipeline generates annotations directly from existing image data. Using standard hardware (A5000 GPUs), the pipeline annotates 558K samples in ~13 hours, demonstrating real-world scalability.
>
> As suggested, we will incorporate the above discussion in the revision.

---

### Official Review · Reviewer_zak5 · 2025-07-02

**Clarity:** 4
**Significance:** 3
**Originality:** 4
**Rating:** 6
**Confidence:** 5

**Summary:**

This paper addresses the issue of fine-grained patch-level alignment of the projector in Multimodal Large Language Models (MLLMs). The paper proposes a novel lightweight alignment loss that improves the token-level semantic matching of visual to textual tokens, and through quantitative and qualitative analysis shows significant improvements to VQA, referring expression, and captioning performance, as well as token-level semantic matching. The paper argues that compression of visual information is critical to the performance of a projector in MLLMs; that while the standard caption loss enables coarse alignment with lower compression, the proposed align loss enables strong compression capability and better patch-level alignment.

**Questions:**

1. Averaging embeddings of tokenized words seems noisy, since the word parts likely carry very different semantic meaning compared to the whole.
   - It would be interesting to understand the effect on the align loss and multi-semantic alignment when it comes to word parts vs whole words.
2. The choice of measuring Von Neumann entropy is deliberate, and Line 141-142 does not clarify why this particular metric was chosen, as opposed to Shannon's entropy, to help quantify the degree of "compression" in visual tokens after projection.
   - It would be helpful for the reader, for instance, to note that Von Neumman metric unlike Shannon's is basis independent, an important consideration as the projector is projecting from the visual to the textual space.

**Ethical Concerns:**

["NO or VERY MINOR ethics concerns only"]

**Final Justification:**

Authors have clearly and strongly clarified all concerns; primarily along the lines of some conflicting results in the paper.

There are no major concerns, and the paper is a technically strong contribution in my opinion.

Many congratulations to the authors on a great contribution.

**Limitations:**

Yes. Addressed.

**Paper Formatting Concerns:**

Nothing major of note.

**Quality:**

4

**Strengths And Weaknesses:**

**Strengths:**

 - **Contributions seem significant and original:**
    - Patch-Level Localization: The proposed align loss is novel and seems to be light-weight, relatively efficient, and performative. Based on both qualitative and quantitative results, the proposed align loss significantly improves the projector's fine-grained visual-text token alignment, with significant improvements across captioning, VQA, and referring expression comprehension. Further, the patch level loss is independent of the LLM's forward pass (requires only the LLM's embedding matrix), and is efficient to compute.
    - Multi-Semantic Alignment: The approach to semantically mapping visual tokens to sets of text embeddings, using the matching pursuit algorithm, is another interesting and novel contribution. This appears to provide, to my knowledge, a first in efficient and effective token-level visual-text semantic matching. The resulting qualitative analysis of mapping the visual tokens to interpretable text tokens is also intriguing, and likely opens up a number of future ideas towards semantic matching analysis in MLLMs.
- **Quality** : The efficacy of the proposed method seems well supported by the qualitative and quantitative results. A number of ablations are provided applying these methods to varying LLMs and projectors, although there is scope for more detailed ablations supporting varying design decisions (see below on ablating beta hyperparameter in align loss).
- **Clarity** : The paper is well written, easy to follow, and provides a good introduction and motivation to the problem and approach. There is some room to strengthen the arguments (see weaknesses below), but a strong write-up overall.


**Weaknessess:**

- **Weak Entropy argument, with potentially contradictory claims**:

  - The observations in Table 1, justifying the claim of projector compression, by using the comparison of Neumann entropy of vision embeddings before/after projection seem unsupported and even contradictory. Specifically, the relatively larger drop in entropy using MLP projectors, compared to linear projectors, is used to justify that "deeper, non-linear transformation can better remove 'redundant' information".  Additionally, this is used to "provide an explanation for the performance advantage of MLP over linear projector". However, C-Abstractor has a higher entropy than both MLPs and linear projectors, and has been reported to be more performative than both, for general visual understanding [1]. This contradicts the reasoning and claim made here.
  - Additionally, the drop in entropy cannot be claimed as being simply due to the removal of "redundant information"; potentially useful visual information may have been lost. Certainly, compression is occurring, and does not detract from the central claims of the paper. However, it is unclear if relating that to performance is well-founded.
  - The same argument appears to be used in Lines 281-282 for Table 3.  It would be helpful to clarify this.


[1] Cha, Junbum, et al. "Honeybee: Locality-enhanced projector for multimodal llm." Proceedings of the IEEE/CVF Conference on Computer Vision and Pattern Recognition. 2024.

- **Need more detailed ablations on design choices**:
    - The choice of linearly increasing beta (hypeparameter for the align loss weight) is not justified or ablated in the paper. In general, it would be helpful to provide more detailed ablations on the choice of the beta hyperparameter, linearly increasing it from 0 to 5 during stage 1.
    - Additionally, it would be helpful to see ablations with other LLM families (beyond LLAMA), as well as other projector types (such as transformer projectors).

---

> ### Author Rebuttal · Authors · 2025-07-30
>
> Thank you for your detailed and positive feedback. We greatly appreciate your recognition of our contributions in patch-level localization and multi-semantic alignment. Below, we respond to the remaining concerns individually and clarify the corresponding points.
>
> > Q1: However, C-Abstractor has a higher entropy than both MLPs and linear projectors. This contradicts the reasoning and claim made here.
>
> A1: Thank you for this observation. In our paper, we primarily compare the Von Neumann Entropy $H(\mathbf{V}\_{after})$ of linear and MLP projectors since they yield similar values when randomly initialized. We don't directly compare with the C-Abstractor because a random initialized C-Abstractor has a much higher baseline entropy ($7.4913$) compared to Linear ($4.8197$) or MLP ($4.8245$). This demonstrates that absolute entropy is influenced by the inductive bias of the model architecture. The C-Abstractor uses a deep CNN-based architecture with residual connections (ResNet blocks), which—even when randomly initialized—produces more diverse outputs. This higher initial entropy reflects the architectural prior of the network, not its training quality.
>
> To fairly assess the projector’s training effect (i.e., information compression caused by learning), we can measure the entropy drop ΔH, which removes the influence of architecture priors: $\Delta H(V) = \frac{H_{random}(V) - H_{trained}(V)}{H_{random}(V)} \in [0,1]$. where we get: $\Delta H_{C-Abs}  = 0.729> \Delta H_{MLP} = 0.578 > \Delta H_{Linear} = 0.485$. This demonstrates that the C-Abstractor achieves higher entropy reduction after training. Thank you for these insightful comments. We will include this discussion with a comprehensive comparison of the C-abstractor in our revision.
>
> > Q2: Additionally, the drop in entropy cannot be claimed as being simply due to the removal of "redundant information"; potentially useful visual information may have been lost. Certainly, compression is occurring, and does not detract from the central claims of the paper.
>
> A2: Thank you for the comments. As briefly discussed in the conclusion, we agree with the reviewer’s perspective that over compression may potentially lead to useful visual information loss. There exists a tradeoff between redundancy removal and semantic loss when the compression level continues to increase. Within a proper compression range, more compression improves performance by removing redundant information and enhancing alignment. However, over-compression causes semantic information loss and performance degradation. To empirically validate this tradeoff, we conducted a controlled ablation where we varied the patch loss weight $\beta$ in the joint objective: $L = L_{caption} + \beta L_{patch}$. Larger $\beta$ encourages stronger patch-level alignment and induces more compression. As shown in the table, performance first improves then drops as $\Delta H$ increases, which validates our conclusion.
>
> | $\beta$ setting                                 | $\beta = 0$ (original LLaVA) | $\beta = 2$ | $\beta = 5$| $\beta = 10$ | $\beta = \infty$(Patch Loss Only) |
> |------------------------------------------|-------------------------|-----------|------------------|------------|--------------------------|
> | $\Delta H$                                        | 0.5789                  | 0.6039    | 0.6646           | 0.6726     | 0.6972                   |
> | Performance (Avg)                        | 58.54                   | 59.78     | **60.56**        | 59.72      | 57.77                    |
>
> In the revision, we will clarify this in section 3 and Lines 281-282 and include these experiments in the appendix.
>
> > Q3: The choice of linearly increasing beta (hypeparameter for the align loss weight) is not justified or ablated in the paper.
>
> A3: Thanks for the suggestions. Due to the limited space, we omit the ablation study in our main paper.  We will now present our ablation studies on the hyperparameter $\beta $ in two aspects:
>
> 1. Linear increasing vs. fixed schedule: The linear schedule gradually increases $\beta $ to impose patch-level alignment progressively. This design choice stabilizes early training and prevents premature over-compression. As shown in the table, when comparing fixed $\beta=5 $ versus linearly increasing $\beta $ from $0$ to $5$, the linear schedule showed better performance. Therefore, we adopted the linear increasing schedule for all subsequent experiments.
>
> 2. Optimal final $\beta $ value: Since $\beta $ in the objective function balances global-level and patch-level alignment, a larger $\beta$ emphasizes patch-alignment loss over caption loss. We examined the impact of $\beta \in \{0, 2, 5, 10\}$, where the baseline LLaVA-1.5 corresponds to $\beta = 0$. As shown in the table, when $\beta $ is too small ($\beta = 0$ or $2$), patch-level alignment remains insufficient, leading to suboptimal performance. Conversely, when $\beta $ is too large ($\beta = 10$), the model overly focuses on local regions, compromising its ability to establish comprehensive correspondence between the entire image and sentences. This imbalance degrades global-level alignment and ultimately harms overall performance.
>
> | Setting                                 | GQA   | Science QA | VizWiz VQA | OKVQA | Avg   |
> |----------------------------------------|-------|------------|------------|-------|-------|
> | fixed ($\beta = 0$)                           | 61.93 | 66.80      | 50.00      | 53.42 | 58.04 |
> | fixed ($\beta = 5$)                           | 62.16 | 68.07      | **54.86**  | 56.51 | 60.40 |
> | linear increasing ($\beta \in [0,2]$)           | 62.64 | 68.12      | 50.84      | 57.52 | 59.78 |
> | linear increasing ($\beta \in [0,10]$)          | 62.64 | 68.57      | 50.82      | 56.85 | 59.72 |
> | linear increasing ($\beta \in [0,5]$) (Optimal)    | **62.99** | **68.67** | 52.29      | **58.29** | **60.56** |
>
> We will include the complete ablation study in the appendix.
>
> > Q4: It would be helpful to see ablations with other LLM families (beyond LLAMA), as well as other projector types (such as transformer projectors).
>
> A4: Thank you for the suggestion. We have ablations on different LLMs (Vicuna and LLAMA) and projector types (linear, MLP, C-Abstractor). Due to time constraints, we were not able to include additional ablations, which we plan to explore in future work. Specifically, we aim to incorporate LLMs with different parameter sizes or architectures such as Phi-3 (3.8B) and Mixtral-8x7B (MoE), and to investigate transformer-based projectors, including Q-Former[1] and D-Abstractor[2].
>
> [1] Li, J., Li, D., Savarese, S., & Hoi, S. (2023, July). Blip-2: Bootstrapping language-image pre-training with frozen image encoders and large language models. In International conference on machine learning (pp. 19730-19742). PMLR.
>
> [2] Cha, J., Kang, W., Mun, J., & Roh, B. (2024). Honeybee: Locality-enhanced projector for multimodal llm. In Proceedings of the IEEE/CVF Conference on Computer Vision and Pattern Recognition (pp. 13817-13827).
>
> > Q5: Averaging embeddings of tokenized words seems noisy, since the word parts likely carry very different semantic meaning compared to the whole. It would be interesting to understand the effect on the align loss and multi-semantic alignment when it comes to word parts vs whole words.
>
> A5: Thank you for pointing this out. Ideally, we would prefer that each word is tokenized into a single unit by the LLM tokenizer. However, in practice, many words—especially longer or less frequent ones—are split into multiple sub-tokens. Let a visual patch embedding be $\mathbf{v}$ and a word *w* be decomposed by the LLM’s tokenizer into $n$ sub‑tokens with embeddings $ [ \mathbf{w}\_{1},  \mathbf{w}\_{2}, …  \mathbf{w}\_{n} ]$. In that case, we aim to maximize total similarity between the patch and **all** sub‑tokens: $\max \sum_{i = 1}^{n} \left <\mathbf{v}, \mathbf{w}\_i \right> $, which is equivalent to $ \max \left <\mathbf{v}, \frac{1}{n}\sum_{i} \mathbf{w}\_i \right >$, i.e., the alignment between the vision token and the average embedding of sub-tokens.
>
> Empirically, Trager et al. [3] show that vision-language models exhibit approximately linear semantic structure, where meanings of complex concepts can be composed by summing component representations. This also supports our choice of averaging sub-token embeddings. That being said, as briefly discussed in the conclusion, we acknowledge that averaging embeddings over sub-tokens might not be the optimal choice and could potentially dilute semantic meaning. If the reviewer has alternative suggestions, we’d be happy to test and incorporate them.
>
> [3] Trager, M., Perera, P., Zancato, L., Achille, A., Bhatia, P., & Soatto, S. (2023). Linear spaces of meanings: compositional structures in vision-language models. In *Proceedings of the IEEE/CVF International Conference on Computer Vision* (pp. 15395-15404).
>
> > Q6: The choice of measuring Von Neumann entropy is deliberate, and Line 141-142 does not clarify why this particular metric was chosen, as opposed to Shannon's entropy, to help quantify the degree of "compression" in visual tokens after projection.
>
> A6: Thank you for the notes. As reviewer suggested, we will provide stronger motivation for our choice of Von Neumann entropy. Von Neumann entropy is calculated from the eigenvalues of the normalized covariance matrix and remains invariant under orthogonal or unitary transformations. This means it captures the intrinsic "spread" of information regardless of the coordinate system used. In contrast, Shannon entropy is defined for discrete distributions or requires discretization in continuous cases. It lacks basis independence—changing the coordinate axes or binning scheme can significantly alter the measured value, creating unwanted variability when comparing embeddings before and after projection. We will clarify this in the final revision.

---

> > ### Comment · Reviewer_zak5 · 2025-08-05
> >
> > I thank the authors for what is an excellent and informative response!
> >
> > Clarifications on Q1/A1 and Q2/A2 are very interesting, and in strong support of the paper's arguments.
> > It would indeed be helpful to add the $\Delta H(V)$ results as well as the patch loss weight ablations in the supplementary for the interested reader.
> >
> > The other concerns were minor, and were clarified well irrespective.
> >
> > Overall, I congratulate the authors on an excellent work. I look forward to seeing this communicated widely at NeurIPS.

---

> > > ### Author Response · Authors · 2025-08-05
> > >
> > > Thank you for your positive and encouraging feedback. We are pleased that our clarifications on Q1/A1 and Q2/A2 were helpful. As suggested, we will include the $\Delta H(V)$ results and the patch loss weight ablation experiments in the appendix in our revision. We sincerely appreciate your thoughtful comments.

---

### Official Review · Reviewer_ZSmz · 2025-07-02

**Clarity:** 3
**Significance:** 3
**Originality:** 2
**Rating:** 4
**Confidence:** 4

**Summary:**

This work treats the projector in multimodal large language models (MLLMs) as having a dual role of “information compression + fine-grained alignment.” First, it uses Von Neumann entropy to quantify and demonstrate that the projector substantially compresses visual redundancy. Next, it introduces Patch-Aligned Training: leveraging PAD data automatically generated by RAM, Grounding DINO, and SAM to apply a cosine alignment loss to each visual patch. Without altering the network architecture, the new projector yields a marked performance boost for LLaVA-7B.

**Questions:**

- This paper relies on a combination of RAM, Grounding DINO, and SAM for automatic annotation. Could you provide evaluation results comparing with manual annotations in terms of IoU/mAP, so as to quantify the true impact of false positives and false negatives on training?

- In the RefCOCO and instruction-following sections, is there a side-by-side comparison with models of the same parameter size?

- Von Neumann Entropy may indicate better alignment, but is there a ΔH versus task performance curve that reveals the tipping point where over-compression causes semantic information loss?

**Ethical Concerns:**

["NO or VERY MINOR ethics concerns only"]

**Final Justification:**

This paper identifies the drawback of projector between LLM and vision encoders. It proposed a multi-semantic alignment hypothesis and patch-aligned training to address the drawback, which achieve good improvement and contribute to the community. The rebuttal address all my concerns. Thus, I recommend to accept this paper.

**Limitations:**

yes

**Quality:**

3

**Strengths And Weaknesses:**

## Strengths
- This paper introduces Patch-Aligned Training based on information compression and fine-grained alignment. By changing only the loss function—with no network modifications—it incurs very low computational overhead yet delivers significant gains on tasks such as localization and VQA.  This paper writing is well written and easy to follow, and performance well.


## Weaknesses
- When Grounding DINO produces erroneous boxes, non-maximum suppression (NMS) and confidence‐score thresholding are employed to remove overlapping or low-quality detections. However, the quality of this filtering has not been evaluated, and the effectiveness of the automatic annotations remains unquantified.

- There is a similar work *SEA: Supervised Embedding Alignment*, which aligns each visual token to the LLM word embedding to solve the problem that caption-loss can only bring coarse-grained semantics.

---

> ### Author Rebuttal · Authors · 2025-07-30
>
> We sincerely thank the reviewer for the feedback. Below, we respond to each of the main concerns individually and clarify the corresponding points.
>
> > Q1: Grounding DINO uses NMS and confidence-score thresholding to remove overlapping or low-quality boxes, but the quality of this filtering and the effectiveness of the automatic annotations have not been evaluated.
>
> A1: Thank you for your suggestion. The Mask-Label Annotation Pipeline serves primarily as an automated method to quickly obtain bounding box and segmentation information from the original LLaVA pretraining dataset to support our patch-aligned training. Since our main contributions are the analysis of multimodal projector and the patch-aligned training method, due to the limited space, we omitted the presentation of evaluation of the Mask-Label Annotation Pipeline in the main paper. We will now present the details of our evaluation.
>
> To optimize our automatic annotation pipeline, we conducted a thorough ablation study using the coco_val_2017 dataset, which provides ground truth bounding boxes. We focused on two key hyperparameters:
>
> 1. Score threshold: Only boxes with confidence scores above this threshold are selected.
> 2. NMS threshold: During non-maximum suppression (NMS), this determines the maximum allowed overlap between two boxes—if their IoU exceeds this threshold, the box with the lower confidence score is removed.
>
> We evaluated performance using F1 score at IoU of 0.5, which classifies predictions as true or false positives based on an IoU threshold of 0.5. First, we tested various score threshold values:
>
> | Score threshold | 0.1   | 0.2   | 0.3   | 0.4   | 0.5   |
> |------------------|-----|--------|--------|--------|--------|
> | F1 @ [IoU=0.5]     | 0.2762 | 0.4931 | 0.6326 | **0.6677** | 0.6207 |
>
> Next, we fixed the score threshold at 0.4 and evaluated various NMS threshold values:
>
> | NMS threshold   | 0.3    | 0.5    | 0.7    | 0.8    | 0.9    |
> |------------------|--------|--------|--------|--------|--------|
> | F1 @ [IoU=0.5]     | 0.6530 | 0.6677 | 0.6702 | **0.6722** | 0.6717 |
>
> Based on this analysis, we selected the optimal hyperparameters (Score threshold = 0.4 and NMS threshold = 0.8) for our final implementation. Following the reviewer's suggestion, in the revision, we will include this evaluation in the appendix.
>
> > Q2: There is a similar work SEA: Supervised Embedding Alignment, which aligns each visual token to the LLM word embedding to solve the problem that caption-loss can only bring coarse-grained semantics.
>
> A2: We thank the reviewer for pointing out the related work. While SEA shares the motivation of improving alignment between visual tokens and language embeddings beyond caption-level supervision, our work differs in several significant ways:
>
> 1. Fine-grained alignment measurement: SEA focuses primarily on introducing a new training loss, whereas our work provides a systematic analysis of existing projectors using novel metrics, including *Von Neumann Entropy* (quantifying information compression) and $\mathrm{Align}(\mathbf{V}, \mathbf{W})$ (measuring fine-grained alignment). These analyses reveal the structural limitations of current projectors and provide actionable insights.
>
> 2. Multi-semantic alignment: SEA treats each patch as corresponding to a single label. In contrast, we introduce a Matching Pursuit-based decomposition to extract multiple aligned tokens per patch, capturing fine-grained multi-token semantics—crucial for complex understanding.
>
> 3. Vocabulary coverage: SEA relies on a fixed, predefined vocabulary, which inherently limits semantic granularity. We use RAM to recognize a broad and dynamic vocabulary of visual concepts, allowing our patch supervision to adapt to diverse real-world entities and contexts.
>
> 4. Loss formulation and performance: While SEA adopts a contrastive loss with handcrafted negative sampling, we use a simpler cosine loss that eliminates the need for sampling strategies. Since SEA doesn’t release their code, we implement the contrastive loss on our own. Our ablation studies show that our approach outperforms contrastive loss in downstream tasks:
> | Dataset     | Original LLaVA | SEA (contrastive loss) | Ours (cos loss) |
> |-------------|----------------|-------------------------|------------------|
> | GQA         | 61.93          | 60.95                   | **62.99**        |
> | Science QA  | 66.8           | 67.58                   | **68.67**        |
> | OKVQA       | 53.42          | 51.59                   | **58.29**        |
>
> We will incorporate these key distinctions and comparisons with SEA in the revision.
>
> > Q3: Could you provide evaluation results comparing with manual annotations in terms of IoU/mAP, so as to quantify the true impact of false positives and false negatives on training?
>
> A3: Thank you for the question. Directly creating manual annotations for the entire LAION dataset would require an impractical amount of human effort and time, so we are unable to provide a direct comparison with fully manual annotations. However, we used the coco-val 2017 dataset, which has high-quality annotated bounding boxes, to evaluate our pipeline with optimal hyper-parameters and compare it with the original Grounding DINO. The results are as follows:
>
> |                      | AP@[IoU=0.50:0.95] | AP@[IoU=0.50] | AP@[IoU=0.75] |
> |----------------------|-------------------|---------------|---------------|
> | Original Grounding DINO | 0.485             | 0.644         | 0.529         |
> | **Our pipeline**         | **0.531**         | **0.676**     | **0.572**     |
>
> AP@[IoU=0.50:0.95] is the mean precision across IoU thresholds from 0.50 to 0.95 (stepped by 0.05). A prediction is considered correct if its overlap with ground-truth exceeds the IoU threshold. As the results demonstrate, our pipeline consistently outperforms the baseline. Following the reviewer's suggestion, we will include this evaluation in the appendix of the revision.
>
> > Q4: In the RefCOCO and instruction-following sections, is there a side-by-side comparison with models of the same parameter size?
>
> A4: Thank you for your question. In both Table 5 (referring expression comprehension) and Table 7 (instruction-following),  our comparisons between Patch-aligned LLaVA (7B) and LLaVA-1.5-7B are conducted using exactly the same models (the same architecture and parameter size (7B)), ensuring a fair evaluation. We additionally compare with LION (Dual-Level vIsual knOwledge eNhanced Multimodal Large Language Model) [1] because it reports zero-shot performance on RefCOCO—without training on RefCOCO—consistent with our evaluation setting. In contrast, many other models (e.g., QwenVL) only report results after fine-tuning on RefCOCO, which leads to higher scores. Although LION uses a larger 12B model, as shown in the table, our 7B model still achieves higher average performance on referring expression grounding tasks, highlighting the effectiveness of our method.
>
> | Model     | RefCOCO val | RefCOCO test A | RefCOCO test B | RefCOCO+ val | RefCOCO+ test A | RefCOCO+ test B | RefCOCOg val | RefCOCOg test  | Avg   |
> |-----------|-------------|--------|--------|---------------|--------|--------|---------------|--------|--------|
> | LION-12B  | 58.54       | 56.41  | **59.36** | 45.93       | 45.73  | 47.89  | **66.12**     | **64.69** | 55.58  |
> | Ours-7B   | **65.97**   | **72.26** | 55.82  | **58.49**   | **66.87** | **48.09** | 55.78       | 56.24 | **59.94** |
>
> [1] Chen, G., Shen, L., Shao, R., Deng, X., & Nie, L. (2024). Lion: Empowering multimodal large language model with dual-level visual knowledge.
>
> > Q5: Von Neumann Entropy may indicate better alignment, but is there a ΔH versus task performance curve that reveals the tipping point where over-compression causes semantic information loss?
>
> A5: Thanks for the question. Yes, there exists a tipping point in $\Delta H$ versus task performance curve. As shown in the table, as $\Delta H$ increases, the overall performance first improves then declines. Before the tipping point, redundant information is removed, which improves performance compared to the original LLaVA with $\beta = 0$. However, after the tipping point, performance drops rapidly as over-compression causes semantic information loss.
>
> | $\beta$ setting                                 | $\beta = 0$ (original LLaVA) | $\beta = 2$ | $\beta = 5$| $\beta = 10$ | $\beta = \infty$(Patch Loss Only) |
> |------------------------------------------|-------------------------|-----------|------------------|------------|--------------------------|
> | $\Delta H$                                        | 0.5789                  | 0.6039    | 0.6646           | 0.6726     | 0.6972                   |
> | Performance (Avg)                        | 58.54                   | 59.78     | **60.56**        | 59.72      | 57.77                    |
>
> Here, the change of entropy is measured as normalized one: $\Delta H(V) = \frac{H_{before}(V) - H_{after}(V)}{H_{before}(V)} \in [0,1]$. Task performance is measured by taking the average of the performance over the QA datasets (GQA, Science QA, VizWiz VQA and OKVQA). The compression level is controlled by the weight of the patch-aligned loss in the aligned training:  $L = L_{caption} + \beta L_{patch}$, where larger $\beta $ leads to larger $\Delta H$, indicating more compression.
>
> Due to space limitations, we omitted this ablation study in the current version. We will include it in the revised version, either in the appendix or in the main paper, depending on the available space.

---

> > ### Author Response · Authors · 2025-08-06
> >
> > Dear Reviewer ZSmz,
> >
> > Thank you again for your time and consideration of our rebuttal. We hope our responses have addressed your concerns satisfactorily. If there are any remaining issues or points that need further clarification, please don’t hesitate to let us know and we would be happy to provide additional explanations. We truly appreciate your feedback and support.

---

> > ### Comment · Reviewer_ZSmz · 2025-08-08
> >
> > Thank authors for the rebuttal. I'm glad to see that all my concerns are carefully addressed with suggested experiments. Will upgrade the rating.

---

> > > ### Author Response · Authors · 2025-08-08
> > >
> > > Thank you for your response and acknowledgment of our rebuttal. We are glad that our rebuttal addressed your comments and sincerely appreciate your constructive feedback.

---

> ### Comment · Area_Chair_7UiB · 2025-08-06
>
> Dear Reviewer ZSmz,
>
> Please take a moment to carefully read the authors’ rebuttal and add your comments.
>
> Thank you for your time and contribution.

---

### Official Review · Reviewer_UhGf · 2025-07-03

**Clarity:** 2
**Significance:** 2
**Originality:** 2
**Rating:** 4
**Confidence:** 4

**Summary:**

This paper tries to enhance the performance of multimodal language models through fine-grained alignment between text and visual cues. It first analyzes the role of projector in compressing vision embeddings and aligning with word embedding. It hence proposes the patch-aligned training, which boosts the performance.

**Questions:**

The authors should further clarify the technical novelty and significance of the proposed method and conclusions.

**Ethical Concerns:**

["NO or VERY MINOR ethics concerns only"]

**Final Justification:**

The rebuttal has addressed part of my concern on the novelty and presentation. I thus would like to raise my score.

**Limitations:**

yes

**Quality:**

3

**Strengths And Weaknesses:**

Strength:
The proposed method boosts the performance of MLLM as shown in experiments.

Weakness:
- The proposed method simply adds a patch loss, which is computed based on automatically generated patch-text pairs. The technical contribution is quite limited.
- The findings in Sec 3 reveals that existing projectors show limited multi-semantic alignment and patch-level alignment. Those findings, however are not quite innovative and can be inferred from the performance of MLLMs. E.g., the poor localization capability indicates the poor patch-level alignment.
- The presentation should be improved. Many symbols are used with different meaning, e.g., V_before, and V_after show different dimensionalities in Sec. 3.1.

---

> ### Author Rebuttal · Authors · 2025-07-30
>
> We sincerely thank the reviewer for the feedback. Below, we respond to each of the main concerns individually and clarify the corresponding points.
>
> > Q1: The proposed method simply adds a patch loss, which is computed based on automatically generated patch-text pairs. The technical contribution is quite limited.
>
> A1: Thank you for your comments. We appreciate the opportunity to clarify our contributions. We want to first note that the patch loss only serves as one component of our contributions. In this work, we develop a comprehensive framework for both analyzing and improving fine-grained vision-text alignment in MLLMs. We highlight our key contributions as follows: (1) examining the projector's role in compressing vision embeddings and aligning them with word embeddings from an entropy perspective, (2) proposing metrics that quantitatively reveal how projectors trained only with caption loss (the standard approach) achieve limited patch-level alignment, resulting in weak and coarse-grained alignment,  (3) using matching pursuit to find the proper word embeddings to preserve multiple semantic meanings,  and (4) finally proposing a patch-alignment method to enhance patch-level alignment.
>
> With respect to the proposed patch loss, as mentioned by Reviewer zak5, it offers several key advantages: (1) Efficient and simplicity. The cosine similarity between visual tokens and LLM’s embedding tokens is computationally efficient. Since the patch-level loss is independent only requires the LLM's embedding matrix and operates independently of the LLM's forward pass, the training time of PatchAligned LLaVA remains comparable to the original LLaVA. (2) Effectiveness. Despite being a straightforward modification, the proposed align loss significantly improves the projector's fine-grained visual-text token alignment, yielding substantial improvements: $16$% on referring expression grounding tasks, $4$% on question-answering tasks, and $3$% on modern instruction-following benchmarks. (3) Adaptable to any MLLM framework. Our approach requires no specialized architecture and can be easily integrated into various MLLMs to enhance fine-grained alignment between different LLM and vision encoders.
>
> > Q2: The findings in Sec 3 reveals that existing projectors show limited multi-semantic alignment and patch-level alignment. Those findings, however are not quite innovative and can be inferred from the performance of MLLMs. E.g., the poor localization capability indicates the poor patch-level alignment.
>
> A2: We respectfully disagree that these findings lack novelty. To the best of our knowledge, no prior work has formally studied patch-level alignment and multi-semantic alignment. If the reviewer is aware of prior work that directly quantifies these alignments, we would be happy to cite it. While poor localization performance of MLLMs may hint at weak patch-level alignment, it can also arise from several other factors, such as limitations of the vision backbone or the language model’s ability to interpret visual tokens. Without a formal metric and analysis, these factors cannot be disentangled. To bridge this gap, Section 3 introduces a quantitative metric ($\mathrm{Align}(\mathbf{V}, \mathbf{W})$) for patch-level alignment and proposes an algorithm to analyze multi-semantic alignment. Using our $\mathrm{Align}(\mathbf{V}, \mathbf{W})$ metric, we show that the original LLaVA projector exhibits weak alignment, improving $\mathrm{Align}(\mathbf{V}, \mathbf{W})$ only from $0.065$ to $0.142$ compared to a random projector. Our proposed  patch-align training further improves $\mathrm{Align}(\mathbf{V}, \mathbf{W})$ to $0.279$, leading to a $16$% gain on referring expression grounding tasks. We will incorporate this point at the beginning of section 3.2.2. in the revision.
>
> > Q3: The presentation should be improved. Many symbols are used with different meaning, e.g., V_before, and V_after show different dimensionalities in Sec. 3.1.
>
> A3: The $\mathbf{V}\_{\mathrm{before}}$ and $\mathbf{V}\_{\mathrm{after}}$ could have different dimensionality. As explained in Eq. (1),  $\mathbf{V}\_{\mathrm{before}}$ and $\mathbf{V}\_{\mathrm{after}}$ refer to features before and after the projector, in MLLMs such as LLaVa which have a structure Vision Encoder $\rightarrow$ Projector $\rightarrow$  LLM. For example, the vision encoder CLIP-ViT-Large-Patch14-336 has an embedding size of $1024$ (the dimension of $\mathbf{V}\_{\mathrm{before}}$), whereas Vicuna 7B has an embedding size of $4096$ (the dimension of $\mathbf{V}\_{\mathrm{after}}$). We will add a clarifying comment after Eq. (1) to make this point explicit. If the reviewer has noticed any other issues with the notation, we would be happy to address them.

---

> > ### Author Response · Authors · 2025-08-06
> >
> > Dear Reviewer UhGf,
> >
> > Thank you again for your time and consideration of our rebuttal. We hope our responses have addressed your concerns satisfactorily. If there are any remaining issues or points that need further clarification, please don’t hesitate to let us know and we would be happy to provide additional explanations. We truly appreciate your feedback and support.

---

> > ### Comment · Reviewer_UhGf · 2025-08-08
> >
> > I have read the feedback from authors, and it has addressed some my concerns. I would like to update my ratings.

---

> > > ### Author Response · Authors · 2025-08-08
> > >
> > > Thank you for your response and acknowledgment of our rebuttal.  We are glad that our rebuttal addressed concerns and we would be happy to address any further questions or suggestions you may have.

---

> ### Comment · Area_Chair_7UiB · 2025-08-06
>
> Dear Reviewer UhGf,
>
> Please take a moment to carefully read the authors’ rebuttal and add your comments.
>
> Thank you for your time and contribution.

---

### Comment · Area_Chair_7UiB · 2025-08-04
**Call for Participation in Author-Reviewer Discussions**

Dear Reviewers,

Thank you for your valuable review - we've identified diverse perspectives that would benefit from your further insights. Kindly review the author response, engage in an open exchange with the authors, and refine your assessment if warranted. Your expertise would be greatly appreciated.

Best regards

---

### Note · Authors · 2025-08-13

We sincerely thank all the reviewers and AC for the constructive feedback and for engaging in the discussion. All reviewers have now acknowledged our responses, with 2 out of 4 reviewers increasing their scores. As a result, there is now consensus among reviewers and no reviewers mentioned any remaining concerns after the rebuttal.

- Reviewer UhGf initially acknowledged our method's improved performance but had concerns about technical contribution due to its simplicity. We clarified our contributions that include both a comprehensive analysis of fine-grained vision-text alignment in MLLMs and method to enhance it, and highlighted our work as the first to formally study patch-level and multi-semantic alignment (to our knowledge) and advantages of our patch loss. The reviewer raised no further concerns and indicated willingness to upgrade the rating.

- Reviewer ZSmz initially acknowledged the performance and simplicity (e.g., "incurs...gains"), but had concerns about the evaluation of our annotation pipeline and the relationship between $\Delta H$ and task performance. We provided additional ablation experiments as suggested, which fully addressed these points. The reviewer appreciated the thorough response and confirmed an upgrade in rating.

- Reviewer zak5 was positive overall, recognizing the analysis (e.g., "a first in efficient...in MLLMs") and the advantages of the proposed method for improving alignment. The main concerns include the discussion of C-Abstractor over-compression. Our rebuttal clarified this issue, and the reviewer agreed it was satisfactorily addressed, raising no further concerns.

- Reviewer 7Lxw was positive overall, recognizing the analysis (e.g., "Strengths:…") and method for improving alignment (e.g, "simple yet effective"). The reviewer requested comparison with similar work, and a deeper discussion of our supervised method. We provided these additional experiments and analysis. The reviewer acknowledged the updates, and we assume no further concerns remain.

For the final version, we will: (1) include the $\Delta H$ results and patch loss weight ablation experiments in the appendix, (2) add a detailed evaluation of our annotation pipeline, and (3) expand the discussion of our contribution and the supervision method and provide comparison to similar work.

We believe the final version of our paper will benefit greatly from these suggestions and will serve as a meaningful contribution to the community’s multimodal alignment.

---

### Decision · Program_Chairs · 2025-09-17

**Decision:**

Accept (poster)

**Comment:**

The paper proposes a detailed analysis of visual-text alignment in MLLMs and introduces a lightweight patch-aligned training objective that improves fine-grained alignment and downstream performance. Across thorough reviews, the consensus is generally positive, with all reviewers ultimately recommending acceptance (1 Strong Accept, 1 Accept, 2 Borderline Accepts).

While some concerns were raised—especially regarding the novelty over existing methods like SEA and the use of automatically generated annotations—these were convincingly addressed in the rebuttal, with additional evaluations and ablations provided.

Given the significance of the fine-grained alignment problem in MLLMs and the empirical rigor demonstrated, I recommend acceptance. The paper provides meaningful insights and a practical solution likely to benefit the community.